

# DTS-AdapSTNet: an adaptive spatiotemporal neural networks for traffic prediction with multi-graph fusion

Wenlong Shi[1], Jing Zhang[1], Xiangxuan Zhong[1], Xiaoping Chen[1] and Xiucai Ye[2]

[1] School of Computer Science and Mathematics, Fujian University of Technology, Fuzhou, China
[2] Department of Computer Science, University of Tsukuba, Tsukuba Science City, Ibaraki, Japan

Corresponding author
Jing Zhang, jing165455@126.com

## ABSTRACT

Traffic prediction is of vital importance in intelligent transportation systems. It enables efficient route planning, congestion avoidance, and reduction of travel time, etc. However, accurate road traffic prediction is challenging due to the complex spatio-temporal dependencies within the traffic network. Establishing and learning spatial dependencies are pivotal for accurate traffic prediction. Unfortunately, many existing methods for capturing spatial dependencies consider only single relationships, disregarding potential temporal and spatial correlations within the traffic network. Moreover, the end-to-end training methods often lack control over the training direction during graph learning. Additionally, existing traffic forecasting methods often fail to integrate multiple traffic data sources effectively, which affects prediction accuracy adversely. In order to capture the spatiotemporal dependencies of the traffic network accurately, a novel traffic prediction framework, Adaptive Spatio-Temporal Graph Neural Network based on Multi-graph Fusion (DTS-AdapSTNet), is proposed. Firstly, in order to better extract the hidden spatial dependencies, a method for fusing multiple factors is designed, which includes the distance relationship, transfer relationship and same-road segment relationship of traffic data. Secondly, an adaptive learning method is proposed, which can control the learning direction of parameters better by the adaptive matrix generation module and traffic prediction module. Thirdly, an improved loss function is designed for training processes and a multi-matrix fusion module is designed to perform weighted fusion of the learned matrices, updating the spatial adjacency matrix continuously, which fuses as much traffic information as possible for more accurate traffic prediction. Finally, experimental results using two large real-world datasets demonstrate that the DTS-AdapSTNet model outperforms other baseline models in terms of mean absolute error (MAE), root mean square error (RMSE), and mean absolute percentage error (MAPE) when forecasting traffic speed one hour ahead. On average, it achieves reductions of 12.4%, 9.8% and 16.1%, respectively. Moreover, the ablation study validates the effectiveness of the individual modules of DTS-AdapSTNet.

# INTRODUCTION

Traffic prediction plays a vital role in intelligent transportation systems. Accurate road traffic forecasting facilitates dynamic route planning, congestion avoidance, travel time reduction, and efficient allocation of traffic resources (*Rabbouch, Saâdaoui & Mraihi, 2018*; *Lana et al., 2018*; *Wang et al., 2022b*). Traffic prediction aims to estimate the future traffic conditions (*e.g.*, traffic flow and speed) for each road segment using historical traffic data. Prediction methods can be categorized into two groups roughly: temporal dependencies-based methods and spatiotemporal dependencies-based methods (*Ren, Li & Liu, 2023*; *Ermagun & Levinson, 2018*).

For the prediction methods that only considering temporal dependencies, such as the Autoregressive Integral Moving Average (ARIMA) model (*Ahmed & Cook, 1979*), and Bayesian model (*Castillo, Menéndez & Sánchez-Cambronero, 2008*). They focus on modeling temporal dependencies of time series mainly without considering potential spatial dependencies among predicted road segments or nodes. However, with the development of deep learning methods, attention has shifted towards considering potential spatial dependencies within traffic networks. In the early stages, the research area is divided into regular grids. The convolutional neural networks (CNNs) and recurrent neural networks (RNNs) are employed to learn spatial relationships and valuable spatiotemporal information among these grids is extracted (*Geng et al., 2019*; *Yu et al., 2017*). Subsequently, with the successful application of graph neural networks (GNNs) in processing graph topology (*Jiang et al., 2023*), Spatio-Temporal Graph Neural Networks (ST-GNNs) have been developed and have demonstrated superior performance compared to grid-based methods (*Yu, Yin & Zhu, 2019*; *Diao et al., 2019*; *Wu et al., 2020*). Compared with previous methods, ST-GNN utilizes predefined graphs, which facilitate more effective learning of latent spatial features. In recent years, the training frameworks of ST-GNN are divided into two parts: the graph learning module and the prediction network module (*Li & Zhu, 2021*; *Lee & Rhee, 2022*). With the advancement of technologies such as public transportation systems and sensors, large amounts of spatio-temporal data can be obtained easily, providing a robust data foundation for traffic prediction (*Liu et al., 2020*; *Jiang & Luo, 2022*). For example, *Wang et al. (2019)* use the historical trajectory data of taxis to extract valuable spatial information through deep neural networks for road traffic prediction. *Ta et al. (2022)* learn the potential spatial relationships among sensors first. They use the historical traffic data provided by the sensors to predict the traffic conditions of each sensor through a spatial–temporal convolutional network.

Unfortunately, road traffic prediction still faces the following three challenges: (1) Graphs based solely on a single spatial relationship may overlook crucial factors such as road characteristics and vehicle flow. This oversight can result in an inaccurate representation of spatial relationships, hindering the extraction of comprehensive spatial dependencies from traffic data. (2) The end-to-end training method leads to interdependence, making it challenging to determine the training direction of the learnable parameters in each module. (3) There are some limitations in the utilization of existing spatio-temporal data, which leads to the inability to extract and fuse the data well. Additionally, as traffic networks

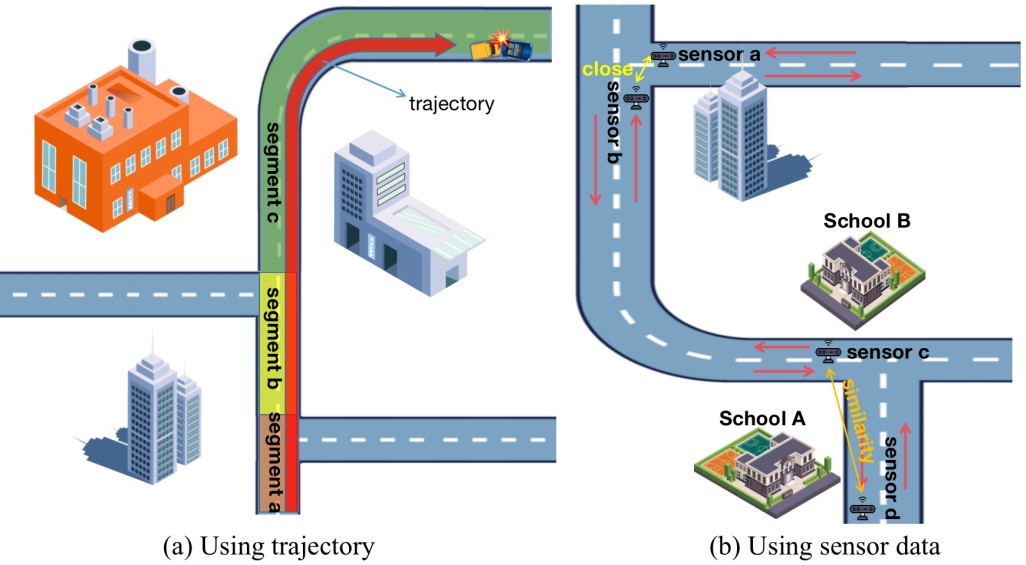

(a) Using trajectory       (b) Using sensor data

**Figure 1** **The drawbacks of using trajectory and sensor data for road traffic prediction.** (A) The long trajectories affected by traffic accidents represent the traffic situation of segments a, b and c roughly. But in fact, segments a and b cannot be represented in such a simple way because they are also connected to other roads. (B) Sensors b and c are far away, and there is no similarity in the environment. Through predefined matrix and adaptive learning, their relationship will be weakened gradually, but in fact, they have a strong correlation as they locate in the same road.

expand, the scalability of prediction models becomes essential. The ability to implement these models efficiently across larger, dynamic environments is vital for their practical application.

An example of the problems is shown in Fig. 1. On one hand, a long trajectory is depicted in Fig. 1A, which represents the traffic conditions of multiple road segments roughly. However, this representation fails to take into account other important traffic details, making it less favorable for road traffic prediction. On the other hand, the distance and the environment are only considered as shown in Fig. 1B, neglecting the flow direction of the actual road and disregarding the relationships among sensors on the same road. Consequently, this simplistic approach also hinders accurate road traffic prediction. Therefore, both representations require improvement to enhance prediction accuracy.

In order to solve the above problems, DTS-AdapSTNet is proposed in this article. Firstly, a novel DTS relationship matrix generation module is designed to address the issue of inaccurate graph predefined by a single spatial relationship. Instead of relying on the Euclidean distance matrix solely, multiple spatial relationship matrices are provided, which are fused to obtain the initial predefined graph. Secondly, a two-stage alternating training structure is proposed to overcome the limitations of traditional end-to-end training. This structure includes alternate training between the adaptive matrix generation module and the prediction module, thereby enhancing control over the training direction of learnable parameters. Finally, sensor data is utilized in experiments to predict the traffic of road

segments more accurately. The topological structure of the entire network is considered when dividing the traffic network into road segments, enabling the identification of corresponding sensors for each segment. Moreover, a well-designed loss function is proposed to train the prediction module, resulting in more accurate road predictions based on sensor data. This process gradually strengthens useful spatial relationships while weakening useless ones, ultimately yielding an optimal spatial adjacency matrix through continuous fusion and updating of learned matrices. The main contributions of this study are summarized as follows:

(1)  A new relationship matrix generation module is proposed. Three kinds of adjacency relationships are calculated among sensors of each road segment. Consequently, the generated adjacency matrices provide a more direct reflection of the relationships among sensors, thereby facilitating the extraction of hidden spatial dependencies among road segments from the traffic data.

(2)  A two-stage alternating training matrix generation method is proposed. Unlike the end-to-end learning method, this method enables better control over the learning direction of parameters. Consequently, it allows for the generation of a learned matrix.

(3)  An improved loss function is proposed for training the prediction module. Additionally, the learned matrices are weighted and fused using a novel weighted fusion mechanism. This training process maximizes the utilization of traffic information data, with the objective of enhancing the training of sensor data for road segment traffic prediction.

(4)  A large number of experiments have been conducted on two real public datasets. The experimental results demonstrate that the proposed model outperforms the comparison models in terms of prediction accuracy.

The rest of the article is organized as follows: 'Related work' reviews existing research related to our work. 'Preliminaries' introduces the motivation of this article, basic definitions and problem description. 'DTS-adapSTNet: An Adaptive Traffic Prediction model with Mulit-graph Fusion' introduces the traffic prediction model proposed in this article. In 'Experiments', experimental evaluations are conducted to evaluate the effectiveness of DTS-AdapSTNet. Finally, the conclusions and future works are presented in 'Conclusion'.

# RELATED WORK

In this section, the relevant methods of existing research are reviewed from three aspects.

## Traditional prediciton methods

One common approach to traffic forecasting involves the application of traditional methods, which utilize statistical or machine learning models applied to historical or real-time traffic data (*Almeida et al., 2022*). Such as linear regression (LR) (*Alam, Farid & Rossetti, 2019*), Autoregressive Integral Moving Average (ARIMA) (*Ahmed & Cook, 1979*), as well as ARIMA-based variant models, *etc.* (*Chen et al., 2011*; *Alghamdi et al., 2019*). With the development of machine learning, several typical machine learning-based models

have emerged, such as random forest regression (RFR) (*Liu & Wu, 2017*), support vector machine (SVM) (*Toan & Truong, 2021*), K-nearest neighbor (KNN) (*Sun et al., 2018*) and Bayesian Network (*AlKheder et al., 2021*), *etc.* However, these methods also face some challenges. Such as dealing with nonlinearity, uncertainty and anomalies in traffic data, requiring a large amount of labeled data, and ignoring the spatial correlations among different locations, *etc.*

In addition, the accuracy of traffic forecasting is not only determined by the chosen prediction methods, but also on the application and processing of various traffic data (*Jeon & Hong, 2016*; *Nagy & Simon, 2021*).

## Prediction methods based on trajectory data

In order to improve applicability to specific problems, researchers often seek to evaluate the performance of predictions on real datasets. The utilization of public transportation trajectory data, which offers convenience and wide coverage, has spurred many researchers to explore the application of deep learning techniques combined with real trajectory datasets for predicting (*Jiang, 2022*). In such approaches, vehicle trajectories (*e.g.*, buses or taxis) are modeled using deep learning techniques to leverage their rich informational content. For example, various deep neural network architectures are employed to learn complex spatiotemporal dependencies between vehicle trajectories and traffic flows.

Initially, some researchers focused on capturing temporal correlations of individual historical traffic data using RNNs, Long short-term memory(LSTM), and Gated Recurrent Units (GRUs) (*Lv et al., 2018*; *Altché & de La Fortelle, 2017*; *Lu et al., 2020*). However, focusing on temporal correlations solely proves insufficient as traffic networks also exhibit complex spatial patterns. Inspired by the success of GNNs and sequence modeling approaches, spatio-temporal GNNs have been introduced to simultaneously capture spatial relationships and temporal dependencies. For instance, *Wang et al. (2022a)* propose a hierarchical traffic flow prediction method based on spatial–temporal Graph Convolutional Networks (GCNs). This method considers the spatial and temporal dependencies of historical trajectories comprehensively, resulting in more accurate traffic flow prediction. However, most existing spatio-temporal GNNs follow the practice of constructing an adjacency matrix based on predefined measurements, such as spatial distance, functional similarity, or traffic connection, *etc.* (*Geng et al., 2019*; *Wang et al., 2019*). This approach involves learning on a predefined adjacency matrix.

Unfortunately, predefined adjacency matrices may not capture spatial relationships adequately or describe them accurately. Recently, dynamic graph generation modules have been adopted in multi-step traffic prediction widely to learn the spatial relationships dynamically. For example, *Djenouri et al. (2023)* propose a spatio-temporal GCNs based on graph optimization to predict urban traffic. A dynamic adjacency matrix is used to reflect the spatial relationships in the traffic network. *Zhang et al. (2022)* also recognize that the traditional end-to-end training methods face challenges in controlling the learning direction of parameters. This limitation results in unclear information from the generated graph and limited improvements in prediction performance. To address this, an alternate training approach is proposed, utilizing historical trajectory flow data of public transportation.

The graph learning module and prediction network are combined, enabling the prediction of future traffic flow for a specific road at a specific time. *Li et al. (2023)* focus on the challenges in integrating and expanding advanced end-to-end spatio-temporal prediction models due to the increasing demands of traffic management and travel planning. To overcome these challenges, they introduce a spatio-temporal pre-training framework that can integrate with downstream baselines to improve performance. This framework includes a spatio-temporal mask auto-encoder with customized parameter learners and a hierarchical spatial pattern encoding network to capture often-neglected spatio-temporal representations and region semantic relationships. Additionally, an adaptive mask strategy is proposed as part of the pre-training mechanism to help the auto-encoder learn robust representations and model different relationships in an easy-to-hard manner.

The above methods can extract the spatio-temporal features in traffic data effectively. However, these methods tend to treat each road as standard grid data or node data, overlooking important details potentially. Consequently, the predicted roads may lack sufficient detail and span a relatively long length. Moreover, these methods rely on the historical trajectories of specific vehicles to predict traffic conditions, which have strong travel characteristics.

## Prediction methods based on sensor data

Road sensor data serves as a significant source of traffic data, enabling the representation of all vehicles and providing more detailed traffic information. Consequently, it serves as an ideal dataset for conducting traffic prediction experiments. In recent years, there has been a parallel advancement in both trajectory-based and sensor-based forecasting methods (*Emami, Sarvi & Bagloee, 2020*). The increasing utilization of sensor data contributes to the development of more general and extensive traffic prediction methods.

*Yu, Yin & Zhu (2017)* introduce a full convolutional structure aimed at capturing spatio-temporal patterns in urban traffic network. *Sun et al. (2020)* develop a multi-view GCNs that captures multiple temporal correlations among sensors from different time intervals. *Li et al. (2018)* reformulate the spatial dependencies of traffic flows as a diffusion process and extend GCNs to directed graphs. *Guo et al. (2019)* propose an attention-based spatio-temporal GCNs to capture deep spatiotemporal correlations among sensors. Recently, graph learning modules have been used to acquire graph structures (*Lee & Rhee, 2022*). *Wu et al. (2019)* combine GCNs with dilated causal convolutional networks to reduce computational costs when processing long sequences. Additionally, an adaptive adjacency matrix is proposed to obtain more reliable spatial correlations among sensors. Considering the highly dynamic nature of urban traffic network, *Ta et al. (2022)* introduce a dynamic GCNs structure for predicting traffic on urban roads. By updating the adaptive adjacency matrix during the training process continuously, the predicted results are more accurate.

The above methods primarily focus on utilizing sensors on the road network as the research subject, resulting in more detailed predictions and demonstrating good performance in predicting individual sensors. However, they overlook the spatial dependencies among sensors on the same road segment. Specifically, the predicted results

tend to prioritize the accuracy of individual sensor predictions rather than predicting traffic for the road segment accurately where the sensors are located. As a result, these methods may not be beneficial for road prediction.

The current forecasting methods often overlook the significance of important spatial dependencies, such as those between sensors on the same road segment, and they provide limited control over the direction of parameter learning during the adaptive learning process. Furthermore, many existing approaches inadequately handle multi-source traffic data, which leads to suboptimal traffic predictions.

To address these limitations, this article proposes a novel traffic prediction framework, DTS-AdapSTNet, which adapts dynamically to spatial relationships within the traffic network. Unlike prior models, our approach generates an adjacency matrix that evolves based on sensor data, learning spatial dependencies between road segments more precisely. By integrating multiple traffic information and utilizing a novel two-stage alternating training structure, the model ensures better control over parameter learning and the prediction process. This enables accurate and adaptive predictions of traffic conditions for specific road segments, overcoming the shortcomings of previous methods.

## PRELIMINARIES

In this section, the motivation, some definitions and the formalization of the problem will be introduced.

### Motivation

The traffic network exhibits not only complex spatial patterns but also constantly changing spatial states. These variations in traffic conditions occur continuously, with each road segment having its own unique driving direction and being equipped with multiple sensors that record traffic data at various time intervals. To achieve accurate traffic prediction, it is essential to analyze large amounts of historical traffic data collected by these sensors. Additionally, precisely grasping the dynamics of traffic network changes is crucial for learning the spatial dependencies more accurately. This grasp is the key to predicting traffic conditions for each road segment at specific future moments.

The proposed DTS-AdapSTNet framework leverages traffic status data collected from sensors to adaptively learn these spatial dependencies. By integrating GCNs, the model enables accurate predictions for individual road segments. The primary goal is to improve road traffic prediction accuracy by dynamically learning the spatial dependencies among sensors in combination with historical traffic data.

### Related definition

**Definition 1** (Crossing sensors). Crossing sensors can be defined as sensors located at both ends of a directed road segment. As shown in Fig. 2A, the sensor at the starting position of a road segment is denoted as $S_i^{in}$, while the sensor at the end position of a road segment is denoted as $S_i^{out}$, where $i$ represents the $i$th road.

**Definition 2** (Sensors on the road segments). Sensors on a road can be defined as all other sensors on a road except crossing sensors, represented as $S_{i,j}^{on}$, where $i$ represents the $i$th road, and $j$ denotes the $j$th sensor on the $i$th road.

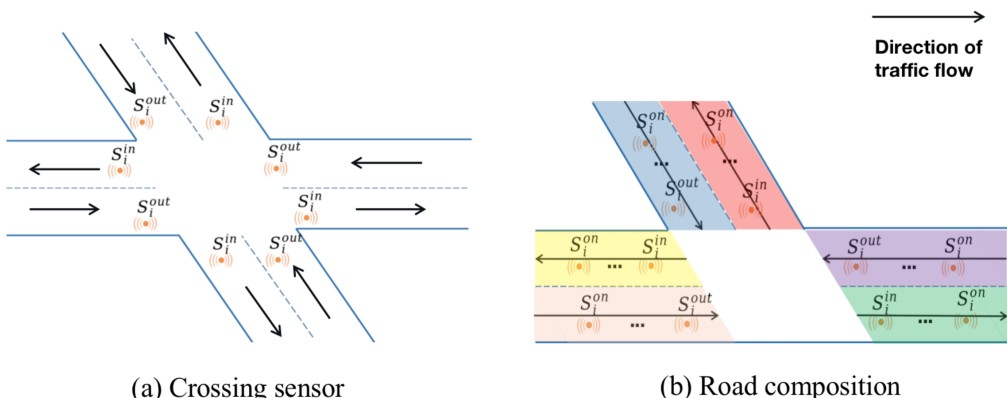

(a) Crossing sensor  (b) Road composition

**Figure 2** **The distribution of sensors on the road.** (A) Definition of starting and ending sensors of road segments. (B) Definition of a road segment and sensors on the road segment.

**Definition 3** (Road segments composition). A road segment $S_i$ in the road network can be defined as composed of an inflow crossing sensor $S_i^{in}$, an outflow crossing sensor $S_i^{out}$, and several sensors $S_{i,j}^{on}$ on the segment.

As shown in Fig. 2B, $S_i = \left\{ S_i^{in}, S_{i,1}^{on}, \ldots, S_{i,j}^{on}, S_i^{out} \right\}$. $S_i$ belongs to the segment set $S$, $S_i^{in}$ and $S_i^{out}$ belong to the crossing sensor set $S^{in-out}$, $\left\{ S_{i,1}^{on}, \ldots, S_{i,j}^{on} \right\}$ belong to the non-crossing sensor set $S^{on}$.

**Definition 4** (Traffic network graph). The traffic network graph can be represented as a weighted directed graph $g = \{S, V, W\}$ of the road network, where $S$ represents the road segment set in the road network and $|S| = N_s$. $V$ represents the set of all sensors and $|V| = N$. $W \in \mathbb{R}^{N \times N}$ is a weighted adjacency matrix representing the spatial correlations of sensors. In general, when $W(i,j) = 0$, it indicates no correlation between sensors $i$ and $j$. However, in this article, some new weighted adjacency matrices among sensors will be defined where this property does not necessarily hold.

## Problem formalization

If the graph signal $X^{(t)}$ represents the historical traffic data observed by each sensor at the $t-$th moment. Then $\mathcal{X} = \left[ X^{(t-T'+1)}, \ldots, X^{(t)} \right]$, $\mathcal{X} \in \mathbb{R}^{T' \times N \times M}$ is used to represent $T'$ historical graphic signals.

**Definition 5** (Traffic prediction problem). The traffic prediction problem aims to use the learned traffic network adjacency matrix $A^* \in \mathbb{R}^{N \times N}$ and $\mathcal{X}$ as the input of the prediction function $g(\cdot)$, so as to predict $T$ future graph signals $\mathcal{Y} = \left[ X^{(t+1)}, \ldots, X^{(t+T)} \right]$, $\mathcal{Y} \in \mathbb{R}^{T \times N \times M}$. As shown in Eq. (1), since $\mathcal{Y}$ represents the traffic conditions predicted in the future. In order to obtain the traffic conditions of each road segment $\tau = \left[ P^{(t+1)}, \ldots, P^{(t+T)} \right]$, $\tau \in \mathbb{R}^{T \times N_s \times M}$, $\mathcal{Y}$ needs to be input into the road segment traffic prediction function, which is shown in Eq. (2).

$$\mathcal{Y} = g\left( \mathcal{X}, A^* \right) \tag{1}$$

**Table 1  Summary of notations encountered frequently.**

| Notation | Description |
|---|---|
| $V$ | Sensor set |
| $S$ | Road segment set |
| $S^{in-out}$ | Crossing sensor set |
| $S^{on}$ | On-road sensor set |
| $S_i^{in} \in S^{in-out}$ | Starting sensor of road $i$ |
| $S_i^{out} \in S^{in-out}$ | End sensor of road $i$ |
| $S_{i,j}^{on} \in S^{on}$ | The $j$th sensor located on road $i$ |
| $S_i \in S$ | The i-th road |
| $N$ | Number of sensors |
| $N_S$ | Number of roads |
| $W \in \mathbb{R}^{N \times N}$ | Adjacency matrix |
| $g = \{S, V, W\}$ | Traffic network graph |
| $M$ | Dimension of node attributes |
| $T', T$ | Window size of measurements |
| $\mathcal{X} \in \mathbb{R}^{T' \times N \times M}$ | Sensor attributes of historical conditions |
| $\mathcal{Y} \in \mathbb{R}^{T \times N \times M}$ | Sensor attributes of predicted conditions |
| $\tau \in \mathbb{R}^{T \times N_S \times M}$ | Road attributes of predicted conditions |
| $A^*$ | Optimal adjacency matrix |
| $g(\cdot)$ | Sensor traffic condition prediction function |
| $h(\cdot)$ | Road traffic condition prediction function |

$$\tau = h(\mathcal{Y}). \tag{2}$$

In general, the size of $X$ can be $\mathbb{R}^{N \times M}$, where $M$ is the number of features observed by each sensor. Similarly, the size of $P$ can be $\mathbb{R}^{N_S \times M}$, where $N_S$ is the number of road segments on the road network, $N_S \leq N . M$ is the number of features observed for each road segment. The datasets used in the experiments only include speed feature, *i.e.*, $M = 1$. However, all results are directly applicable to problems with $M > 1$. A summary of key notations used in our model is shown in Table 1.

## DTS-ADAPSTNET: AN ADAPTIVE TRAFFIC PREDICTION MODEL WITH MULIT-GRAPH FUSION

### Architecture of DTS-AdapSTNet

The overall framework of DTS-AdapSTNet is shown in Fig. 3, which consists of five steps:
**Step 1:** DTS Relationship Matrix Generation module (DTSRMG-module). The position and distance data of each sensor on the road network are put into the DTSRMG-module to generate the distance relationship matrix, transfer relationship matrix and same-road relationship matrix, respectively.
**Step 2:** Adjacency Matrix Predefined module (AMP-module). The three matrices obtained in Step 1 are put into the matrix set $A$. They are fused through the AMP-module to obtain the predefined adjacency matrix $A^*$.

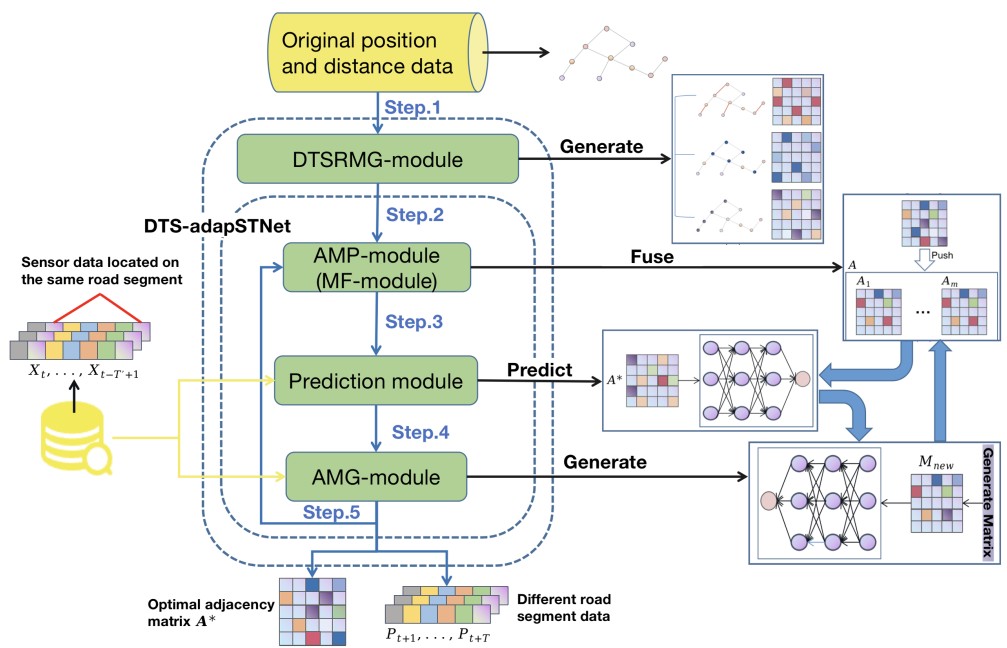

**Figure 3** The overall framework of DTS-AdapSTNet.

**Step 3:** Prediction module based on improved loss function. The relationship matrix $A^*$ is utilized as the optimal input for the prediction module, while incorporating the historical traffic data to train the prediction network simultaneously. A well-designed loss function is employed to facilitate optimization. The parameter $\theta^*$ that maximizes the likelihood estimation of the prediction model's excellence is determined under the condition of the current relationship matrix $A^*$.

**Step 4:** Adaptive Matrix Generation module (AMG-module). The parameter value of the prediction network in the AMG-module is designated as $\theta^*$, which is obtained in Step 3 and remains fixed. The same prediction network is utilized for training, thereby allowing the AMG-module to generate the relationship matrix repeatedly. As a result, a new relationship matrix $M_{new}$ that improves the prediction outcome under the current parameter $\theta^*$ can be obtained. The spatial dependencies among sensors are thus reweighted by this matrix, leading to more accurate predictions ultimately.

**Step 5:** Multi-matrix Fusion module (MF-module). The $M_{new}$ generated in Step 4 is incorporated into the matrix set $A$ of the MF-module, which utilizes the matrices generated in Step 1 as its elements. Through the MF-module, weight calculation and distribution are performed on all matrices in the matrix set. Subsequently, the optimal adjacency matrix is obtained by fusing with the new weights, and it replaces the original $A^*$.

By repeating steps 3-5, a relatively better spatial adjacency matrix and a relatively better road segment traffic prediction model can be obtained as the preset maximum number of training iterations is reached. The pseudocode for DTS-AdapSTNet is described in Algorithm 1.

---

**Algorithm 1:** The implementation of DTS-AdapSTNet

    **Input:** original road network information $R$, parameter set $P$, constant set $C$, itera-
        tions $I$

    **Output:** a model with good prediction performance DTS-AdapSTNet, the optimal
        adjacency matrix $A^*$

1  Execute **Procedure 1** *DTS Relationship Matrix Generation*:

2  $W_D, W_T, W_S \leftarrow$ Generate three adjacency matrix according to different dependen-
   cies ;

3  Execute **Procedure 2** *Generation of initial input matrix*:

4  $A^* \leftarrow$ Fusing three adjacency matrices for initialization;

5  **for** $i$ to $I$ **do**

6      Execute **Procedure 3** *Traffic condition prediction*:

7      $\theta \leftarrow$ Get the best training parameter through training;

8      Execute **Procedure 4** *Adaptive Matrix Generation*:

9      $A_{new} \leftarrow$ Get a new matrix through training;

10     Execute **Procedure 5** *Multi-matrix Fusion*:

11     $A^* \leftarrow$ Get the next iteration of input matrix through filtering and fusion;

12 **end**

13 **return** $A^*$

---

## DTS relationship matrix generation module

Traffic prediction is of great importance in intelligent transportation systems, but accurate road traffic prediction faces challenges due to the complex spatiotemporal dependencies in the traffic network. Existing methods have shortcomings in capturing spatial dependencies. They often consider only a single spatial relationship while ignoring crucial factors such as road characteristics and vehicle flow. This results in inaccurate descriptions of spatial relationships and difficulty in extracting comprehensive spatial dependencies, which affects the accuracy of traffic prediction (*Shuman et al., 2013*). In order to capture the spatial correlations in the traffic network more accurately and enhance the accuracy of road segment traffic prediction, multigraphs are employed in this module to conceptualize the complex traffic network. Three traffic network graphs are defined, namely Distance relationship, Transfer relationship, and Same-road relationship. Combining these three spatial relationships can better reflect the spatial connection between sensors and improve the accuracy of prediction results.

### Distance relationship

A certain correlation is observed among sensors, with the strength of correlation being stronger among sensors that are relatively close to each other. To describe this relationships, the measure of distance among sensors is utilized. The distance between two sensors $i$ and $j$ in the road network, denoted as $dist(i,j)$, is considered as the shortest distance when one or more paths exist between them. This is illustrated in Fig. 4A.

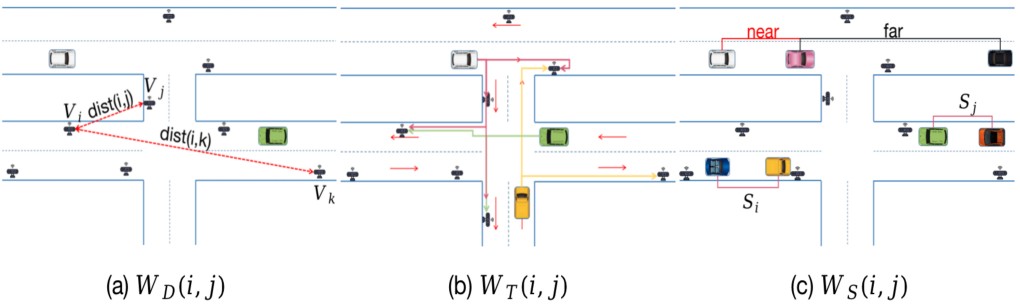

(a) $W_D(i,j)$       (b) $W_T(i,j)$       (c) $W_S(i,j)$

**Figure 4** **Three types of weighted adjacency matrices.** (A) Sensor distance relationship. (B) Transfer status of vehicles. (C) Relationship of the same road segment.

**Definition 6** (Distance relationship matrix $W_D$). The distance relationship matrix $W_D$ is defined by Laplace kernel function (*Paclık et al., 2000*), which is shown in Eq. (3).

$$W_D(i,j) = \begin{cases} e^{(-\|\text{dist}(i,j)\|/\theta_1)}, & i \neq j \text{ and } \text{dist}(i,j) > k \\ 0, & \text{otherwise} \end{cases} \tag{3}$$

where $\theta_1$ is a fixed parameter and $k$ is a threshold. Although the Gaussian kernel function has become the standard for most distance modeling, in theory, a deep neural network can model any function according to the general approximation theorem. Additionally, since the provided experimental data includes the distance among sensors, the Laplacian kernel function is used to calculate $W_D$. By comparing the result calculated by Gaussian kernel function, it can be observed that the former ensures greater accuracy in the experimental outcome.

*Transfer relationship*

The transfer relationship can be used to describe the flow relationship among various road segments. As shown in Fig. 4B, multiple possibilities exist for the transfer situation of each crossing in the network. Accurately simulating the spatial transfer relationship among crossing sensors is of great importance for predicting purposes. To address this, the transfer relationship matrix $W_T$ is defined, which is utilized to simulate the spatial transfer relationship of the road network.

**Definition 7** (Transfer relationship matrix $W_T$). The spatial dependencies between the crossing sensors $S_i^{in}$ and $S_i^{out}$ are captured through the transfer relationship. Consequently, a relationship matrix is obtained which reflects the similarity among crossing sensors effectively.

Firstly, the similarity matrix among crossing sensors is calculated using the Node2Vec algorithm (*Grover & Leskovec, 2016*). Given that the traffic network is a directed and unweighted graph, the crossing sensors are sampled through a biased random walk, which is shown in Eq. (4).

$$P(c_i = x | c_{i-1} = v) = \begin{cases} \dfrac{\pi_{vx}}{Z}, & \text{if } (v,x) \in E \\ 0, & \text{otherwise} \end{cases} \tag{4}$$

where the left side of the equation represents the probability of moving from the current crossing node $v$ to the next crossing node $x$, while $E$ represents the set of all successor neighbor crossing nodes of the current crossing node $v$. $Z$ is a normalization constant, typically the total number of nodes.

Secondly, the transition probability is calculated by Eq. (5).

$$\Pi_{vx} = \alpha_{pq}(t,x) \cdot W_{vx} \tag{5}$$

$$\alpha_{pq}(t,x) = \begin{cases} \dfrac{1}{p}, & \text{if } d_{tx} = 0 \\ 1 & \text{if } d_{tx} = 1 \\ \dfrac{1}{q} & \text{if } d_{tx} = 2 \end{cases} \tag{6}$$

where $t$ represents the previous node, $W_{vx}$ represents the weight of the edges in the weighted graph. Since the traffic network is a directed and unweighted graph, the value of $W_{vx}$ can be regarded as 1. Additionally, $\alpha_{pq}(t,x)$ is defined as the meta-transition probability. Its calculation formula is shown in Eq. (6), where $p$, $q$ are the parameters that control the model walking strategy. $d_{tx} = 0$ indicates that the flow back from the current crossing node, *i.e.*, $t = x$, and $d_{tx} = 1$ indicates that $t$ and $x$ are connected directly, which is the so-called breadth-first walk. $d_{tx} = 2$ means that $t$ and $x$ are not connected, which is the so-called depth-first walk.

The retrograde situation is not feasible in the real road segments. In addition, the breadth-first walking strategy can better capture the dependencies between each crossing node and its directly adjacent nodes. Therefore, when setting the model parameters, the value of $p$ is set to infinity, while the value of $q$ is set as a number greater than 1, making the model more inclined towards the breadth-first walk strategy.

Finally, the vector representation of all nodes $E_{node} \in \mathbb{R}^{N \times C_{node}}$ is obtained through the aforementioned random walk strategy, where $C_{node}$ represents the information dimension recorded by each node. The similarity matrix, denoted as $W_T$, is then calculated based on the customized threshold $T_{sim}$ for transfer relationships. This matrix represents the transfer relationships and is computed by Eqs. (7)–(8).

$$W_T(i,j) = \frac{E_{\text{node}}(i) \cdot E_{\text{node}}(j)}{\|E_{\text{node}}(i)\| \|E_{\text{node}}(j)\|} \tag{7}$$

$$W_T(i,j) = \begin{cases} 0, & W_T(i,j) < T_{sim} \\ 1, & W_T(i,j) \geq T_{sim} \end{cases} \tag{8}$$

where $W_T(i,j)$ represents the similarity matrix of any two crossing sensor vectors. In order to make the transfer relationship matrix $W_T$ sparse further, it is reassigned according to the threshold to obtain the final $W_T$.

*Same-road relationship*

In the traffic network of urban roads, there are closer spatial dependencies among sensors located on the same road. Describing such spatial dependencies more accurately is crucial for predicting road segment conditions. Unlike previous works where the road network is simplified to a grid structure, this study provides a more detailed description of the road network. In reality, most roads consist of two or more lanes, making it inappropriate to classify them as a single road simply. Additionally, distances among sensors on the same road need to be considered. As shown in Fig. 4C, the spatial dependencies among them are also different.

**Definition 8** (Same-road relationship matrix $W_S$). In order to capture the far and near spatial dependencies of sensors located on the same road segment accurately, a deep learning model is not employed directly for learning. Instead, the spatial relationship matrix indicates the relationships among sensors in individual lanes explicitly, including the near, mid, and far relationships.

Firstly, the Euclidean distance is calculated among sensors on each lane according to the original spatial information to obtain the distance relationship matrix $W_S$.

Secondly, a set of fine-grained filters, denoted as $\{f_m\}$, is defined to be applied to $W_S$, $f_m(W_S) = W_S^m \in \{W_S^1, \ldots, W_S^M\}$. The function $f_m(\cdot)$ can be perceived as a filter function, with a total of $M$ such functions available. This implies that $W_S$ is divided into $M$ fine-grained divisions, and the resulting partition matrices satisfy the condition $\sum_{m=1}^{M} f_m(W_S) = 1$. This condition ensures that $W_S$ is distributed across $M$ fine-grained partition matrices, while preserving the distribution characteristics of $W_S$.

Finally, the $m$-th fine-grained partition matrix represents the probability that the relationship between any two sensors in the network belongs to the $m$-th relationship. To facilitate this, a weight set $W = \{w_1, \ldots, w_m\}$ is designed. By combining the probability matrices and weight set $W$, a relationship matrix can be constructed to quantify the level of mutual influence between sensors within each segment. The value of $M$ can be determined based on cluster analysis. To ensure smooth treatment of boundary values, a Gaussian kernel filter is selected. The degree of mutual influence among sensors can be defined by the matrix $W_S$, which is shown in Eqs. (9)–(11).

$$G_m(W_s) = e^{-\frac{(W_s(i,j) - r_m)^2}{2\theta_2^2}}, m \in M \tag{9}$$

$$f_m(W_s) = \frac{G_m(W_s)}{\sum_{m=1}^{M} G_m(W_s)}, m \in M \tag{10}$$

$$W_s = \sum_{m=1}^{M} f_m(W_s) \cdot w_m, m \in M, w_m \in W \tag{11}$$

where $G_m(W_s)$ represents the result obtained through the Gaussian kernel filter, $r_m$ is the cluster center point obtained through cluster analysis, and $\theta_2$ is the hyperparameter.

Three fine-grained partition functions, namely, near, middle, and far, are defined to process $W_s$ through cluster analysis. Additionally, three weight matrices, $W\{w_{near}, w_{mid}, w_{far}\}$, are designed specifically to correspond with these partition functions. It is ensured that $w_{near} > w_{mid} > w_{far}$. The specific process of generating the DTS relationship matrix is shown in Algorithm 1.

## Adjacency matrix predefined module

The performance of the prediction module and the AMG-module significantly depends on the quality of input matrix's initialization. Inaccurate or unreliable initial associations among sensors can lead to suboptimal optimization of the prediction module, subsequently compromising the overall performance of the entire model. Therefore, it is essential to ensure the accuracy and reliability of the initial input matrix to enhance the model's predictive capability and overall performance.

Therefore, this article does not use a single relationship matrix as the initial spatial relationship matrix, nor does it generate the initial matrix randomly. Instead, multiple matrices generated by the DTSRMG-module are used to construct the initial matrix, which includes different types of spatial dependencies. The initial input matrix A can be initialized by Eqs. (12)–(13).

$$
\begin{aligned}
&\text{i.} \quad A_D = \tilde{D}_D^{-\frac{1}{2}} \tilde{W}_D \tilde{D}_D^{-\frac{1}{2}} \\
&\text{ii.} \quad A_T = \tilde{D}_T^{-\frac{1}{2}} \tilde{W}_T \tilde{D}_T^{-\frac{1}{2}} \\
&\text{iii.} \quad A_S = \tilde{D}_S^{-\frac{1}{2}} \tilde{W}_S \tilde{D}_S^{-\frac{1}{2}}
\end{aligned}
\tag{12}
$$

$$
A(i,j) = \frac{A_D(i,j) + A_T(i,j) + A_S(i,j)}{b[A_D(i,j)] + b[A_T(i,j)] + b[A_S(i,j)]}
\tag{13}
$$

where $\tilde{W}_k = W_k + I_N$, $\tilde{D}_k^{(i,j)} = \sum_j \tilde{W}_k^{(i,j)}$, $k$ represents $D, T$ and $S$. Equation (12) is a renormalization technique proposed by *Kipf & Welling (2019)*, which ensures the comparability of different matrices. $b[\cdot]$ represents a binarization function, which is defined by Eq. (14). The specific process of initialization is shown in Algorithm 2.

$$
b[x] = \begin{cases} 1 & if \quad x \neq 0 \\ 0 & else \end{cases}.
\tag{14}
$$

## Prediction module based on improved loss function

Graph convolution-based spatio-temporal neural network is an approach for traffic prediction. It takes one or more adjacency matrices and historical time series data as input. The aim is to capture the spatio-temporal features that are hidden in the historical data, which is shown in Fig. 5A, The network consists of several spatio-temporal blocks typically. Each block captures the spatial dependencies of sensors in the road network after information aggregation using a GCN model. Simultaneously, a temporal attention module or GRUs is combined to capture long sequence dependencies and obtain temporal dependencies of traffic data.

---

**Algorithm 1: Procedure 1:** *DTS Relationship Matrix Generation*

---

**Input:** original road network information $R$, hyper-parameters $k, \theta_1, \theta_2, p, q, T_{sim}$, weight set $W$

**Output:** relationship matrix $W_D, W_T, W_S$

1   Initialize $W_D, W_T, W_S, dist$ as zeros matrixs, $M$ as a empty set;

2   // Method of generating $W_D$

3   $dist \leftarrow$ Get all distance values among sensors from $R$;

4   **for** $dist(i,j)$ **in** $dist$ **do**

5      **if** $dist(i,j) > k$ **then**

6         $W_D \leftarrow e^{(-\|dist(i,j)\|/\theta_1)}$;

7      **else**

8         $W_D \leftarrow 0$;

9      **end**

10   **end**

11   // Method of generating $W_T$

12   $N_C \leftarrow$ Get all number of crossing sensors from $R$;

13   $G \leftarrow$ Create a graph based on the crossing sensors information from $R$;

14   $E_{node} \leftarrow$ Get vector representation of crossing sensors based on Node2vec$(G, p, q)$;

15   **for** $i$ **in** $N_C$ **do**

16      **for** $j$ **in** $N_C$ **do**

17         $W_T(i,j) \leftarrow E_{node}(i) \cdot E_{node}(j)/(\|E_{node}(i)\| \|E_{node}(j)\|)$;

18         **if** $W_T(i,j) \geq T_{sim}$ **then**

19            $W_T(i,j) \leftarrow 1$;

20         **else**

21            $W_T(i,j) \leftarrow 0$;

22         **end**

23      **end**

24   **end**

25   // Method of generating $W_S$

26   $W_S \leftarrow$ Get the distance relationship matrix of each sensor on each road segment;

27   $M \leftarrow$ Get a set of cluster center values by conducting cluster analysis on $W_S$;

28   **for** $r_i, i$ **in** $M, size(M)$ **do**

29      $G_i(W_S) \leftarrow e^{((W_S - r_i)^2/2\theta_2^2)}$;

30      $G_{all} \leftarrow G_{all} + G_i(W_S)$;

31   **end**

32   **for** $r_i, w_i, i$ **in** $M, W, size(M)$ **do**

33      $f_i(W_S) \leftarrow G_i(W_S)/G_{all}$;

34      $W_S \leftarrow W_S + f_i(W_S) \cdot w_i$;

35   **end**

36   **return** $W_D, W_T, W_S$

---

---

**Algorithm 2: Procedure 2:** *Generation of initial input matrix*

---

**Input:** $W_D, W_T, W_S$, a binaryzation function $b[\cdot]$

**Output:** predefined adjacency matrix $A^*$

1   Initialize $W, A_{set}$ as a empty set, $I_N$ as a identity matrix of size $N \times N$, $A$ as a zero matrix of size $N \times N$;

2   **Push** $W_D, W_T, W_S$ **into** $W$;

3   **for** $W_k$ **in** $W$ **do**

4      $\tilde{W}_k \leftarrow W_k + I_N$;

5      **for** $i$ to $N$ **do**

6         $\tilde{D}_k^{(i,j)} \leftarrow$ Add all the columns corresponding to row $i$ of $\tilde{W}_k$;

7      **end**

8      $A_k \leftarrow \tilde{D}_k^{-\frac{1}{2}} \tilde{W}_k \tilde{D}_k^{-\frac{1}{2}}$ ;

9      **Push** $A_k$ **into** $A_{set}$ ;

10   **end**

11   $A_D, A_T, A_S \leftarrow A_{set}[0], A_{set}[1], A_{set}[2]$;

12   **for** $i$ to $N$ **do**

13      **for** $j$ to $N$ **do**

14         $A^*(i,j) \leftarrow (A_D(i,j) + A_T(i,j) + A_S(i,j)) / (b[A_D(i,j)] + b[A_T(i,j)] + b[A_S(i,j)])$;

15      **end**

16   **end**

17   **return** $A^*$

---

For all the models used in this article, each layer of the GCN employs spectral graph convolution based on the Chebyshev polynomial approximation (ChebConv). As shown in Fig. 5B, it is able to capture the spatial dependencies, which can be expressed by Eq. (15).

$$H^{(l)} = ChebConv(A^*, H^{(l-1)}, \theta^{(l)}) = \sigma\left(\sum_{k=0}^{K} T_k(A^*) H^{(l-1)} \theta^{(l)}\right) \tag{15}$$

where $\sigma$ is the activation function, and $T_k(x)$ represents the recursively defined Chebyshev polynomial given by $T_k(x) = 2x T_{k-1}(x) - T_{k-2}(x)$, where $T_0(x) = 1$, $T_1(x) = x$. $A^* = \tilde{D}^{-\frac{1}{2}} \tilde{A} \tilde{D}^{-\frac{1}{2}}$ is the normalized adjacency matrix, where $\tilde{A} = A + I$. $\tilde{D}$ is a diagonal matrix, $\tilde{D}_{ii} = \sum_j \tilde{A}_{ij}$. $H^{(l)}$ represents the hidden features of layer $l$ obtained after the convolution operation, and $\theta^{(l)}$ represents the learnable parameter corresponding to the $l$-th layer structure.

For the training of the previous prediction modules, the focus is usually on designing a loss function to enhance the accuracy of predicting the value of individual sensors, rather than prioritizing the accurate prediction of traffic conditions on the road segment where the sensor is located. This approach may result in less accurate predictions. This article aims to predict the future traffic conditions of each road segment. Instead of calculating the loss function between the real value $Y$ and the predicted value *Pred* of a single sensor directly, some improvements should be made to ensure that the designed loss function optimizes the model's performance in predicting road traffic conditions.

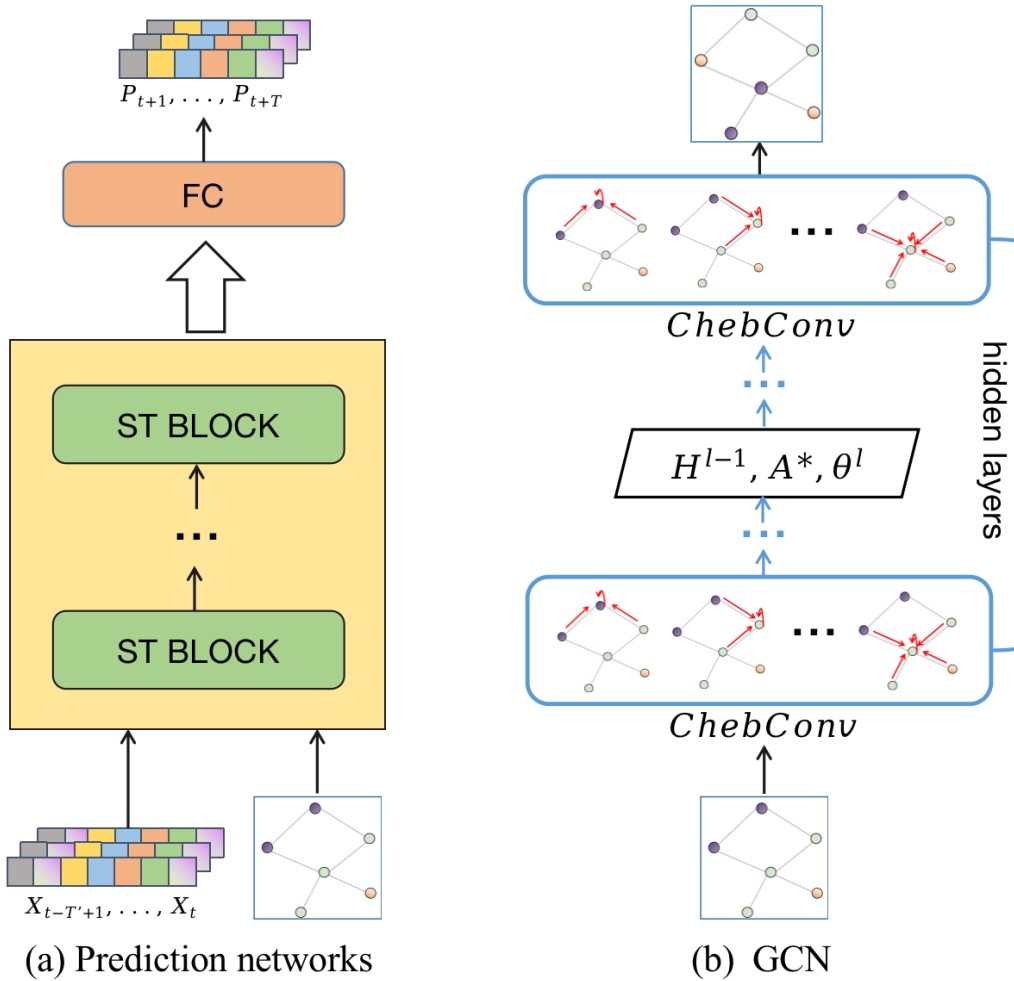

(a) Prediction networks        (b) GCN

**Figure 5** **The general structure of GCN-based prediction neural networks for traffic prediction.** (A) The structure of prediction. (B) The structure of GCN.

Firstly, a matrix $W_r$ of size $N \times S$ should be generated. This matrix is obtained by considering the distribution of sensors in the road segments. The number of sensors in the network is denoted as $N$, while the number of road segments is represented by $S$. The matrix $W_r$ can be defined by Eq. (16).

$$W_r(i,j) = \begin{cases} 1 & V_i \ locates \ on \ S_j \\ 0 & otherwise \end{cases} \tag{16}$$

where $V_i \in N$ represents the $i$th sensor, and $S_j \in S$ represents the $j$th road. If the $i$th sensor locates on the $j$th road, the corresponding matrix value is set to 1, otherwise it is 0. This grouping allows sensors to be categorized according to the road they belong to.

Secondly, the ground truth $Y$ and predicted value $Pred$ are processed. As shown in Eqs. (17)–(18), the actual value $Y_{road}$ of each road segment and the predicted value $Pred_{road}$ are

obtained.

$$Y_{road} = Y \cdot W_r \tag{17}$$

$$Pred_{road} = Pred \cdot W_r. \tag{18}$$

An array $C$ of length $S$ is calculated to record the number of sensors for each road segment. Additional processing is conducted on $Y_{road}$ and $Pred_{road}$. The average real value $Y_{road}$ and the average predicted value $Pred_{road}$ are obtained by Eqs. (19)–(20).

$$Y_{road} = Y_{road}/C \tag{19}$$

$$Pred_{road} = Pred_{road}/C. \tag{20}$$

Finally, the minimization of the $L1$ loss between the predicted value of the road segments and the real value is selected as the training goal of the prediction module, which is shown in Eq. (21).

$$L_p(Y_{road}, Pred_{road}) = |Y_{road} - Pred_{road}|. \tag{21}$$

Through such a loss function, after several iterations, correlations that are favorable for road segment traffic prediction are highlighted, while weaker correlations are erased gradually. The prediction modules of the experiments are all implemented based on several GCNs models with good prediction effects. The specific process is shown in Algorithm 3.

---

**Algorithm 3: Procedure 3:** Traffic condition prediction

**Input:** time series data of traffic feature $X$, target data $Y$, normalized adjacent matrix $A^*$, sensor distribution matrix $W_r$, array $C$, GCN-based spatiotemporal neural networks for traffic prediction $STnet(\cdot)$, number of epoch $N_{epoch}$.

**Output:** well-trained parameter $\theta$ of the prediction module

1 **for** $i$ to $N_{epoch}$ **do**
2     $pred \leftarrow STnet(X, A^*, \theta)$ ;
3     $Y_{road} \leftarrow Y \cdot W_r/C$ ;
4     $pred_{road} \leftarrow pred \cdot W_r/C$ ;
5     Compute $L_p(pred_{road}, Y_{road})$ ;
6     Update model parameters $\theta$ through Adam;
7 **end**
8 **return** $\theta$

---

## Adaptive matrix generation module

In the context of traffic prediction, accurately representing the spatial dependencies among sensors is crucial for obtaining accurate predictions. However, traditional end-to-end training methods often face challenges in determining the training direction of learnable parameters, resulting in unclear information from the generated graph and limited improvements in prediction performance.

To address these limitations, the AMG-module is introduced. It can generate matrices based on the DTSRMG-module, which provides initial spatial dependencies. By utilizing the training results of the prediction network and incorporating prior knowledge in the iterative process, the AMG-module can continuously generate new spatial dependency matrices. This allows the matrices to better capture and represent the stronger dependencies among sensors, thereby improving the accuracy of the model's predictions.

Firstly, an initial relationship matrix is generated according to the cosine similarity by Eqs. (22)–(23).

$$sim(i,j) = \frac{v_i \cdot v_j}{\|v_i\| \|v_j\|} \tag{22}$$

$$A_{ij} = \begin{cases} 1, & if \ sim(i,j) > 0 \\ 0, & otherwise \end{cases} \tag{23}$$

where $v_i$, $v_j$ are the eigenvectors of sensors $i$ and $j$, respectively. $\|\cdot\|$ represents the modulus of the vector. $A_{ij}$ represents the degree of correlation between sensors $i$ and $j$. Then a learnable matrix $A_1$ is introduced and combined with the initial relationship matrix $A$ by Eq. (24) to obtain the initial learnable matrix $M_{init}$.

$$M_{init} = ReLU(A + A_1). \tag{24}$$

$A_1 \in \mathbb{R}^{N \times N}$ are learnable parameters. In order to enhance the sparsity of $M_{init}$, the activation function $ReLU$ can set the diagonal position of the matrix and the other half positions to 0.

Secondly, an attention mechanism is used to fuse the old matrices with the newly generated spatial dependency matrix. Then, an attention weight $\alpha_{ij}$ between each pair of nodes can be obtained by Eq. (25).

$$\alpha_{ij} = softmax(LeakyReLU(W \cdot [v_i \| v_j])) \tag{25}$$

where $W$ is a learnable weight matrix, $v_i$ and $v_j$ are eigenvectors of nodes $i$ and $j$ respectively, $\|$ represents vector concatenation operation, $LeakyReLU$ is a linear rectification function with leakage, and the *softmax* function is used to normalize attention weights. The fusion matrix $M_{fs}$ is obtained according to the attention weights $\alpha_{ij}$ by Eq. (26).

$$M_{fs}(i,j) = \alpha_{ij} \cdot M_{init} + (1 - \alpha_{ij}) \cdot M_{old} \tag{26}$$

where $M_{old}$ is the old spatial dependency matrix.

Finally, to facilitate calculation, the fused matrix is adjusted. In this process, certain elements with relatively small values are filtered out, and control is exerted through the utilization of a hyperparameter $\sigma$. Furthermore, by employing the renormalization technique (*Kipf & Welling, 2019*), a newly generated matrix $M_{new}$ is obtained by Eqs. (27)–(28).

$$M'_{fs} = ReLU(D_{fs}^{-\frac{1}{2}} M_{fs} D_{fs}^{-\frac{1}{2}} - \sigma) \tag{27}$$

$$M_{new} = (D'_{fs})^{-\frac{1}{2}} M'_{fs} (D'_{fs})^{-\frac{1}{2}} \tag{28}$$

where $D_{fs}, D'_{fs}$ are diagonal matrices, $D_{fs}^{(i,j)} = \sum_{j=1}^{N} M_{fs}^{(i,j)}, (D'_{fs})^{(i,j)} = \sum_{j=1}^{N} (M'_{fs})^{(i,j)}. \sigma \in (0,1)$ is a custom threshold, the specific process is shown in Algorithm 4.

---

**Algorithm 4: Procedure 4:** Adaptive Matrix Generation

---

**Input:** time series data of traffic feature $X$, target data $Y$, normalized adjacent matrix $A^*$, sensor distribution matrixa $W_r$, array $C$, GCN-based spatiotemporal neural networks for traffic prediction $STnet(\cdot)$, the set of eigenvector of each node $v$, the parameters $A_1 \in R^N$, $W$, $\sigma$, number of epoch $N_{epoch}$

**Output:** adaptive generator matrix $M_{new}$

1  $M_{old} \leftarrow A^*$;

2  **for** $i$ to $N_{epoch}$ **do**

3      $M_{new} \leftarrow$ Generate a new matrix $G(M_{old}, v, A_1, W, \sigma)$;

4      $pred \leftarrow STnet(X, M_{new}, \theta)$;

5      $Y_{road} \leftarrow Y \cdot W_r / C$;

6      $pred_{road} \leftarrow pred \cdot W_r / C$;

7      Compute $L_p(pred_{road}, Y_{road})$;

8      Fix $\theta$, update model parameters $A_1, W, \sigma$ through Adam;

9  **end**

10  **return** $M_{new}$

---

## Multi-matrix fusion module

After the AMG-module, a new adjacency matrix is generated. This adjacency matrix is then input into the MF-module. For the $N_m$ matrices in this module, they are fused to generate the input matrix $A^*$ for the next iteration of the prediction module. The main task of this module is to weight the importance of different subgraphs through the prediction loss of the prediction model. By doing so, it can identify the most relevant and useful subgraphs for the prediction task. Subsequently, these weights are used to fuse different subgraphs to obtain the input $A^*$ for the next iteration of training. This ensures that the model can make full use of the information from different subgraphs, enhancing the accuracy and reliability of the prediction.

Firstly, the prediction module is utilized to calculate the corresponding prediction loss for all matrices in the matrix set $A$, which is shown in Eq. (29).

$$L_k = L_p(STnet(X, A_k, \theta) \cdot W_r / C, Y_{road}) \tag{29}$$

where $STnet(\cdot)$ is the prediction network. $\theta$ is the parameter that makes the best prediction currently. $k$ refers to the $k$-th submatrix, $W_r$ and $C$ are the matrix and array mentioned in Prediction module. The predicted and actual values of the road are calculated to obtain the predicted loss of the $k$-th submatrix. These prediction loss values are combined into a vector $\vec{l} = (L_1, L_2, ..., L_m)^T$, and then the maximum value is take as $L_{max} = max(\vec{l})$. Then,

it is necessary to judge whether the current matrix number $m$ is greater than the maximum capacity $N_{max}$. If it exceeds, it is necessary to remove the matrix with the largest predicted loss value in $\vec{l}$, which is shown in Eq. (30). Additionally, the maximum value $L_{max}$ is updated.

$$\vec{l} = remove(L_{max}).$$ (30)

---

**Algorithm 5: Procedure 5:** Multi-matrix Fusion

**Input:** time series data of traffic feature $X$, target data $Y$, matrix set $A$, optimized
   parameter $\theta$, the capacity $N_{max}$ of $A$

**Output:** fusion matrix $A^*$

1  $m \leftarrow$ Get the number of items of $A$ ;
2  **for** $i$ **to** $m$ **do**
3  |   $L_k \leftarrow$ Calculate the predicted loss value for each submatrix ;
4  |   **push** $L_k$ **into** $\vec{l}$ ;
5  **end**
6  **if** $m > N_{max}$ **then**
7  |   Remove the maximum predicted loss value from $\vec{l}$ ;
8  |   Remove the corresponding matrix from $A$ ;
9  |   $m \leftarrow$ Get the number of items of $A$ ;
10 **end**
11 $L_{mean} \leftarrow mean(\vec{l})$ ;
12 $w \leftarrow g(\vec{l} - L_{mean})$ ;
13 **for** $i$ **to** $m$ **do**
14 |   $A^* \leftarrow A^* + w_i A_i$ ;
15 **end**
16 **return** $A^*$

---

Secondly, the weight vector $\vec{w}$ of all matrices remaining in $A$ is calculated by Eq. (31).

$$\vec{w} = g(\vec{l} - mean(\vec{l}))$$ (31)

where $mean(\vec{l})$ represents the average of the prediction loss of all matrices. $g$ represents the normalization function, which is defined by Eq. (32).

$$g(\vec{x}) = \left| \frac{e^{\vec{x}}}{\sum_{i=1}^{m} e^{\vec{x}}} \right|$$ (32)

Finally, the input matrix of the next iteration can be obtained through the MF-module, which is shown in Eq. (33).

$$A^* = \sum_{k=1}^{m} w_k A_k$$ (33)

The same normalization is performed on $A^*$ with Eqs. (12)–(13). The specific process of MF-module is shown in Algorithm 5.

**Table 2  The statistics of METR-LA and PEMS-BAY.**

| Datasets | Nodes | Edges | Time windows | Statistical characteristics | Time period covered |
|---|---|---|---|---|---|
| METR-LA | 207 | 1515 | 17568 | traffic speed | 2012.3.1-2012.4.30 |
| PEMS-BAY | 325 | 2369 | 52116 | traffic speed | 2017.1.1-2017.6.30 |

# EXPERIMENTS

In this section, the effectiveness of the DTS-AdapSTNet is evaluated and compared with other baseline models using two real-world datasets. In addition, ablation experiments are also conducted on the relevant modules of the proposed model.

## Datasets

In order to verify the performance of the model, experiments are conducted on two public datasets, METR-LA and PEMS-BAY. The detailed statistics of the two datasets are shown in Table 2.

METR-LA: This traffic dataset contains traffic information collected from loop detectors in the highway of Los Angeles County (*Gehrke et al., 2014*). Since there are a large number of missing values in this dataset and to facilitate the experiment, traffic speed statistics data from March 1, 2012, to April 30, 2012, encompassing 207 sensors along Los Angeles County highways is utilized to mitigate the impact of missing values on experimental results. Moreover, missing values are handled.

PEMS-BAY: This traffic dataset is collected by California Transportation Agencies (CalTrans) Performance Measurement System (PeMS). We selected six months of traffic speed statistics from January 1, 2017 to June 30, 2017, covering 325 sensors in the Bay Area.

The same data pre-processing procedures are adopted as *Li et al. (2018)*. Observations from the sensors are aggregated into 5-minute windows. The original spatial dependencies such as road distance and direction are used as the input of the DTSRMG-module to generate corresponding relationship matrices. The input data are normalized using Z-score. Both datasets are split by time sequence with 70%, 15% and 15% for training, validation and testing, respectively.

## Baseline

Three predictive neural networks based on GCN with good performance are employed as prediction models for experiments. The structures of these three GCN-based frameworks are shown in Fig. 6.

(1) ASTGCN (*Guo et al., 2019*): Attention Based Spatio-Temporal Graph Convolutional Network, which combines graph convolutions with spatial attention to capture spatial patterns, and leverages standard convolutions and temporal attention to extract temporal features.

(2) TGCN (*Zhao et al., 2020*): Temporal Graph Convolutional Network model, which utilizes recurrent models to extract temporal features and graph convolutions to capture spatial dependencies, respectively.

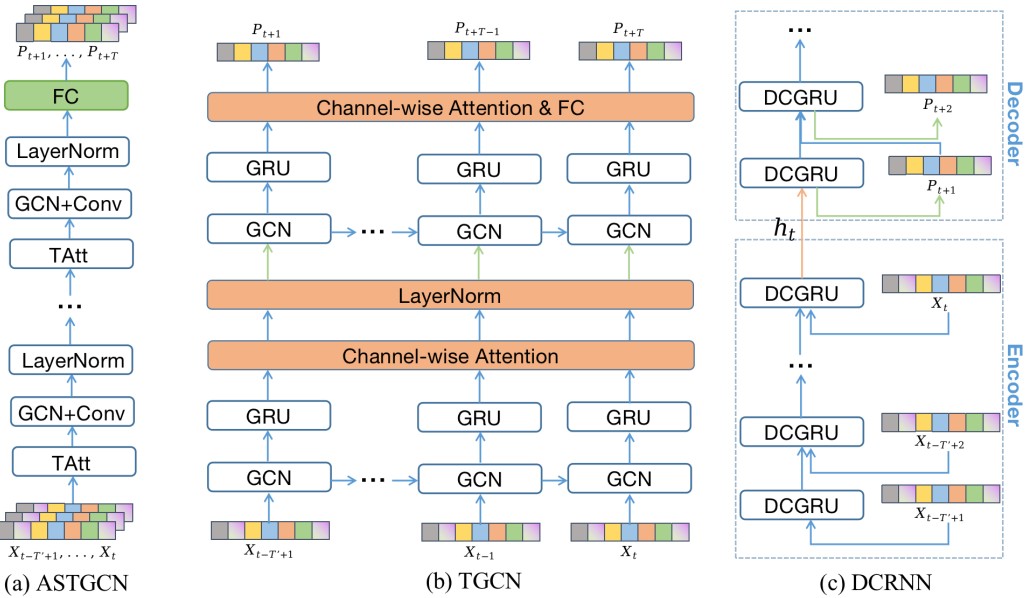

**Figure 6** **The structures of three GCN-based frameworks for multi-step traffic prediction.** (A) The architecture of ASTGCN, featuring the temporal attention module (TAtt) and a combined block (GCN+Conv) that extracts high-dimensional features through both graph convolution and standard convolution. (B) The architecture of TGCN, which includes a Channel-wise Attention module that assesses the significance of various features derived from the GRU. (C) The architecture of DCRNN, where the DCGRU module merges graph convolutional techniques with the GRU framework.

(3) DCRNN (*Li et al., 2018*): Diffused Convolutional Recurrent Neural Network, which replaces matrix multiplication in recurrent models (*i.e.,* GRU and LSTM) with graph convolutions, and extracts spatial and temporal features in an encoder–decoder manner simultaneously.

The above three models also serve as baseline models, along with the following ones, including both classic methods and state-of-the-art approaches:

(1) HA: Historical average, which uses the average of historical traffic flow data to complete the task.

(2) STGCN (*Yu, Yin & Zhu, 2017*): Spatio-Temporal Graph Convolutional Network, which combines 1D convolution and graph convolution.

(3) Graph WaveNet (*Wu et al., 2019*): A convolutional network architecture, which introduces adaptive graphs to capture hidden spatial dependencies and uses dilated convolutions to capture temporal dependencies.

(4) DCRNN+AdaGL (*Zhang et al., 2022*): An Adaptive Graph Learning Algorithm for Traffic Prediction Based on Spatiotemporal Neural Networks, which combines the proposed adaptive graph learning module with a DCRNN to find the adjacency matrix relations that makes traffic prediction work well.

### Evaluation metrics

Three common metrics for traffic prediction are employed to evaluate the performance of different models, including MAE, RMSE and MAPE. Their definitions are shown in Eqs. (34)–(36).

(1) MAE: Mean absolute error measures the average magnitude of errors in the predictions without considering their direction. This makes it a straightforward metric for evaluating the accuracy of predictions in real-world applications like traffic prediction, where absolute deviations matter.

$$\text{MAE}(Y, \widehat{Y}) = \frac{1}{|\Omega|} \sum_{i \in \Omega} |y_i - \widehat{y}_i| \tag{34}$$

(2) RMSE: Root mean square error is particularly sensitive to larger errors, which can be critical in traffic forecasting scenarios where large deviations from actual values could have significant real-world consequences (*e.g.*, incorrect predictions during peak traffic hours).

$$\text{RMSE}(Y, \widehat{Y}) = \sqrt{\frac{1}{|\Omega|} \sum_{i \in \Omega} (y_i - \widehat{y}_i)^2}. \tag{35}$$

(3) MAPE: Mean absolute percentage error provides a normalized error measurement, making it useful when comparing prediction performance across datasets with different scales (*e.g.*, high-traffic *vs.* low-traffic areas).

$$\text{MAPE}(Y, \widehat{Y}) = \frac{1}{|\Omega|} \sum_{i \in \Omega} \left| \frac{y_i - \widehat{y}_i}{y_i} \right| \times 100\%. \tag{36}$$

### Experiment settings

In all experiments, the traffic speed is predicted over the next hour using the traffic speed from the previous hour, hence $T = 12$. The parameters $\theta$, $k$, and $T_{sim}$ used in generating the three relationship matrices $W_D$, $W_T$, $W_S$ are adjusted according to the scale of the data. For the prediction module, the number of training epochs $N_{epoch}$ is set to approximately 10 based on the convergence rate of the prediction module. In the AMG-module, the dimensions of $A_1$ and $A_2$ are configured as 64, and $\sigma$ is selected from the range [0,1]. Regarding the MF-module, the size of the matrix set $A$ is set to 3.

All experiments are executed under a platform with NVIDIA GeForce RTX3080-10GB graphics card. For all deep learning based models, the training process is implemented in Python with Pytorch 1.8.0. Adam (*Wang, Xiao & Cao, 2022*) is utilized as the optimization method with a learning rate of 0.001. Additionally, an early stopping strategy is employed to determine whether the stopping criterion is met. Furthermore, the optimal parameters are determined based on the performance on the validation dataset.

### Performance analysis

Table 3 presents the performance comparison of different baselines and DTS-AdapSTNet on the two datasets for road speed prediction in the next 15 min, 30 min and 60 min, respectively. The prediction networks, which serve as the foundations for DTS-AdapSTNet,

**Table 3  Performance comparison of different methods on METR-LA and PEMS-BAY for traffic speed prediction.**

| Datasets | Model name | 15 min | | | 30 min | | | 60 min | | |
|---|---|---|---|---|---|---|---|---|---|---|
| | | MAE | RMSE | MAPE (%) | MAE | RMSE | MAPE (%) | MAE | RMSE | MAPE (%) |
| METR-LA | HA | 4.67 | 9.37 | 13.8 | 4.67 | 9.37 | 13.8 | 4.67 | 9.37 | 13.8 |
| | ST-GCN | 3.49 | 6.48 | 8.5 | 4.43 | 8.19 | 9.5 | 6.06 | 10.41 | 13.7 |
| | ASTGCN | 2.85 | 6.04 | 7.1 | 3.62 | 7.82 | 9.7 | 4.81 | 10.05 | 13.7 |
| | TGCN | 2.81 | 5.89 | 7.0 | 3.54 | 7.62 | 9.6 | 4.67 | 9.75 | 13.4 |
| | DCRNN | 2.76 | 5.87 | 6.8 | 3.49 | 7.61 | 9.3 | 4.62 | 9.71 | 13.2 |
| | Graph WaveNet | 3.16 | 6.38 | 8.6 | 3.54 | 7.35 | 9.8 | 4.52 | 9.93 | 13.3 |
| | AdapGL+DCRNN | 2.78 | 5.83 | 6.5 | 3.38 | 7.41 | 9.6 | 4.44 | 9.74 | 13.7 |
| | DTS-AdapSTNet(ASTGCN) | 2.73 | 5.77 | 6.6 | 3.39 | 7.39 | 9.0 | 4.37 | 9.37 | 12.4 |
| | DTS-AdapSTNet(TGCN) | 2.76 | 5.76 | **6.3** | 3.37 | 7.31 | 8.7 | 4.30 | 9.28 | 12.4 |
| | DTS-AdapSTNet(DCRNN) | **2.66** | **5.73** | 6.4 | **3.27** | **7.28** | **8.6** | **4.13** | **9.11** | **11.5** |
| PEMS-BAY | HA | 2.90 | 6.15 | 6.9 | 2.90 | 6.15 | 6.9 | 2.90 | 6.15 | 6.9 |
| | ST-GCN | 1.91 | 3.59 | 4.7 | 2.35 | 4.71 | 5.7 | 3.11 | 6.02 | 7.3 |
| | ASTGCN | 1.53 | 2.81 | 3.7 | 1.90 | 3.98 | 4.8 | 2.48 | 5.37 | 6.4 |
| | TGCN | 1.36 | 2.70 | 2.6 | 1.78 | 3.86 | 3.8 | 2.37 | 5.24 | 5.4 |
| | DCRNN | 1.35 | 2.69 | 2.8 | 1.78 | 3.85 | 3.9 | 2.37 | 5.27 | 5.3 |
| | Graph WaveNet | 1.32 | 2.71 | 2.8 | 1.78 | 3.87 | 3.7 | 2.24 | 5.09 | 5.2 |
| | AdapGL+DCRNN | 1.35 | 2.68 | 2.8 | 1.79 | 3.85 | 4.0 | 2.22 | 4.77 | 5.5 |
| | DTS-AdapSTNet(ASTGCN) | 1.34 | 2.65 | 2.7 | 1.77 | 3.70 | 3.9 | 2.26 | 4.76 | 5.2 |
| | DTS-AdapSTNet(TGCN) | 1.33 | 2.62 | 2.7 | 1.76 | 3.68 | 3.9 | 2.22 | **4.70** | 5.0 |
| | DTS-AdapSTNet(DCRNN) | **1.30** | **2.59** | **2.6** | **1.73** | **3.64** | **3.7** | **2.21** | 4.72 | **4.9** |

**Notes.**
The bold values are used to highlight which model has the best performance within each evaluation metric across different datasets.

are denoted within brackets (*e.g.*, DTS-AdapSTNet (ASTGCN) refers to using ASTGCN as a prediction network). The results demonstrate that DTS-AdapSTNet achieves excellent results in all metrics across the prediction range. Superior performance is observed for DTS-AdapSTNet based on three different GCN prediction networks compared to the other baselines, for both the METR-LA and PEMS-BAY datasets. Particularly, DTS-AdapSTNet based on DCRNN exhibits the best performance. When compared with the traditional model HA for a 15-minute prediction on both datasets, it is found that MAE, RMSE, and MAPE are reduced by 43%, 39%, and 54%, respectively. This demonstrates the significance of considering spatial dependencies among sensors. Other spatio-temporal GCN models are also compared, revealing an average reduction in prediction error for each metric by 10%, 6% and 13%, respectively. This reduction can be attributed to the accuracy of the learned adjacency matrix, which has a relatively large impact on prediction results. It is evident from the results presented in Table 3 that as the prediction window increases, the accuracy of prediction decreases for each model. However, DTS-AdapSTNet outperforms other models consistently, with a more noticeable improvement.

For the MAE, when the prediction window is 15 min, the DTS-AdapSTNet (DCRNN) model exhibits a 24% lower error compared to the ST-GCN model. With a prediction window of 30 min, the error is reduced by 26%. Furthermore, with a prediction window of

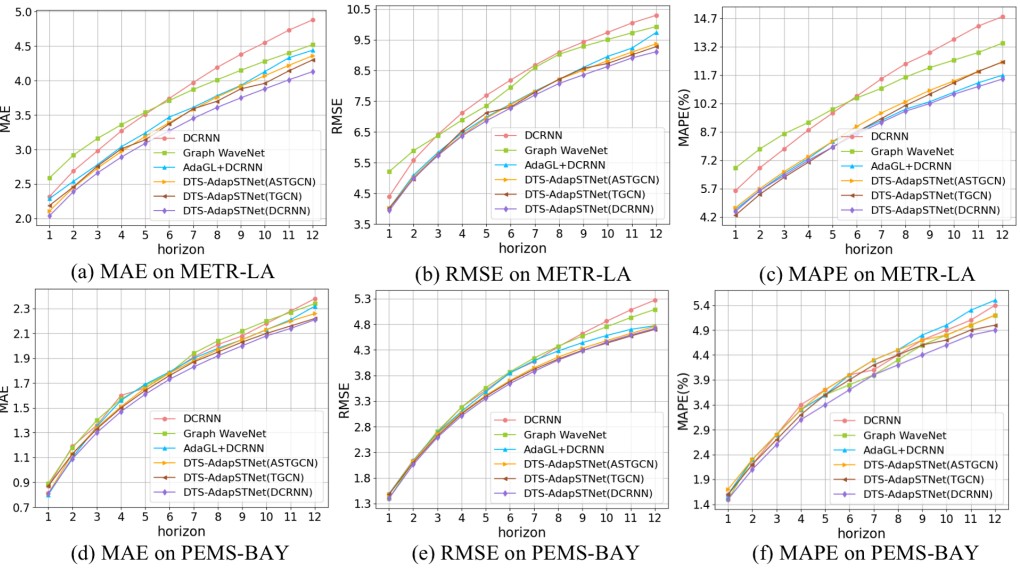

**Figure 7** **Performance comparison of the tested models at each horizon on METR-LA and PEMS-BAY datasets for traffic speed prediction, where one horizon denotes 5 min.** (A), (D) Comparison of MAE. (B), (E) Comparison of RMSE. (C), (F) Comparison of MAPE.

60 min, the error is reduced by 32%. This showcases the increasing complexity of the traffic network and the growing difficulty in prediction as the prediction range expands. However, it is noteworthy that DTS-AdapSTNet is able to outperform in long-term forecasting as well, highlighting the stability of the method employed in this study.

When compared with the AdapGL+DCRNN model, the method in this article has different prediction focuses and specific implementation methods. In different GCN prediction networks, the performance of the DTS-AdapSTNet model has been improved. The most obvious improvement is observed in DTS-AdapSTNet (DCRNN). Particularly noteworthy is the significant decline observed in all three metrics for the METR-LA dataset. Similarly, for the PEMS-BAY dataset, DTS-AdapSTNet continues to remain competitive or even show further enhancements compared to the AdapGL+DCRNN model.

In order to provide a more intuitive comparison of the performance of DTS-AdapSTNet and other models with superior performance under different prediction windows, a line chart illustrating the changes in each evaluation metric corresponding to different models under various prediction windows is depicted in Fig. 7. It can be observed that DTS-AdapSTNet based on different GCNs demonstrates relatively good performance in each prediction range on the two datasets. Specifically, on the METR-LA dataset, as the prediction window increases, the MAE, RMSE and MAPE of the three models proposed in this article increase among 0.32−2.15, 1.00−5.26 and 1.07−7.60, respectively. The corresponding increases of the other three baseline models are 0.32−2.21, 0.98−5.45 and 1.07−7.63, respectively. It can be observed that the error rise of the DTS-AdapSTNet model changes less as the prediction window increases, illustrating the effectiveness of the proposed model in long-term prediction further.

In order to compare the prediction effectiveness of DTS-AdapSTNet with baselines, the comparison between the prediction curves of different models for the speed of the same road on the same day and the ground truth is displayed intuitively in Fig. 8A. It can be observed that during periods of drastic speed fluctuations (*i.e.,* 6:00-12:00), the prediction curve of Graph WaveNet does not fit well with the curve of the ground truth, failing to capture abrupt changes accurately. The curves of AdapGL+DCRNN and DTS-AdapSTNet (DCRNN) align more closely with the actual change shape of the ground truth, depicting the beginning and end of peak hours accurately. However, a considerable deviation from the ground truth is noticeable with AdapGL+DCRNN during the 6:00-7:00 period in the morning, and a significant deviation from the ground truth speed value is observed at 9:00 as well. In contrast, the curve of DTS-AdapSTNet (DCRNN) aligns with the curve of the ground truth better. Similar conclusions can be drawn from the analysis of Fig. 8B. This demonstrates that although DTS-AdapSTNet and AdapGL+DCRNN have a similar structure, the method designed by DTS-AdapSTNet is more conducive to extracting potential spatial relationships between sensors within each road segment. It exhibits a stronger modeling ability for complex and changeable traffic conditions.

## Ablation experiment

To verify the effectiveness of the main modules proposed in this article, ablation studies are conducted on METR-LA and PEMS-BAY datasets.

### Effect of DTSRMG-module and AMG-module

In order to verify the influence of the DTSRMG-module and AMG-module on the experimental results, three variants of each model are tested on two datasets. The results displayed in Fig. 9 indicate that whether it is the METR-LA dataset or the PEMS-BAY dataset, the performance of each model significantly deteriorates when one of the two modules is removed, particularly evident in the METR-LA dataset. As depicted in Figs. 9A–9C, when the DTSRMG-module is eliminated, there is an average increase of 8.9%, 6.1%, and 10.7% in the MAE, RMSE, and MAPE for each model, respectively.

When the AMG-module is removed, there is an average increase of 6.8%, 6.2%, and 8.8% in the corresponding three metrics, respectively. This demonstrates that removing the DTSRMG-module has a larger negative impact on the performance of most models compared to removing the AMG-module. This highlights the vital role of the DTSRMG-module in the predefined relationship matrix. Without the DTSRMG-module, the effectiveness of the AMG-module cannot be realized fully. Similar findings are observed in the analysis of the PEMS-BAY dataset, which are shown in Figs. 9D–9F.

### Effect of generator matrix

The effectiveness of the generated matrix can be observed more intuitively by utilizing DTS-AdapSTNet (DCRNN) for experiments on the METR-LA dataset, where 40 sensors in the dataset are selected. The heat map shown in Fig. 10 depicts the distance relationship matrix, initialization matrix and generation matrix, respectively. Consequently, it is evident from the previous experiments that the experimental prediction results can be improved significantly through the combination of the generation matrix and the DTSRMG-module.

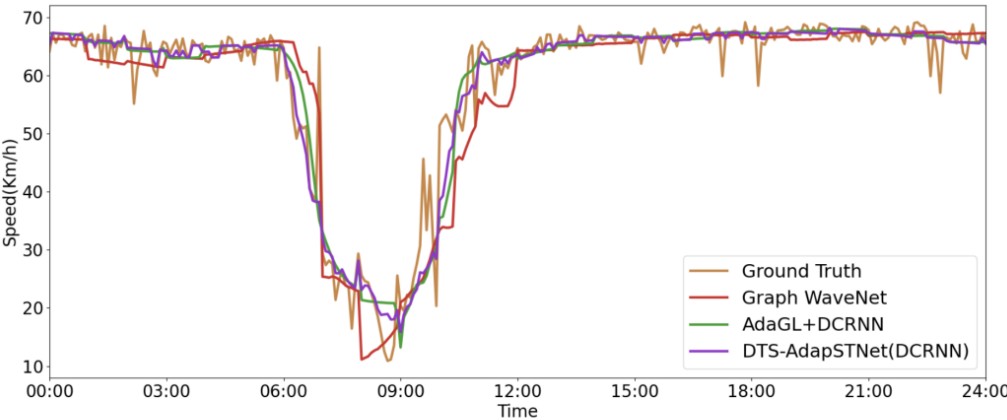

(a) Prediction curves on METR-LA

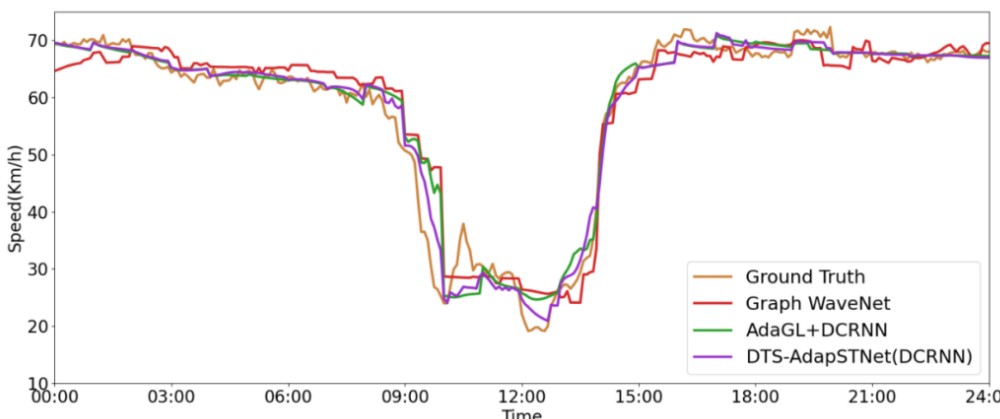

(b) Prediction curves on PEMS-BAY

**Figure 8** **Prediction curves change within a day.** (A) Comparison of prediction curves for one hour ahead prediction on a road of the test data of METR-LA. (B) Comparison of prediction curves for one hour ahead prediction on a road of the test data of PEMS-BAY.

The position distributions of the sensors 160, 164, 161 are shown in Fig. 10E. Firstly, the distance relationship is shown as the small squares 1 and 2 in Fig. 10A, where square 1 represents the relationship between sensors 160 and 164, and square 2 represents the relationship between sensors 164 and 161. It is evident that the distance between them is related closely. Secondly, after the initialization matrix is generated by the AMP-module, the spatial relationships among them are shown in Fig. 10B. The color of squares 1 and 2 becomes lighter, indicating that the relationships among sensors at this stage not only consider the distance factor but also take into account other factors. Finally, after the AMG-module, the spatial relationships among the three sensors are shown in Fig. 10C. It can be seen that the color of square 1 is darker, indicating that after the model training, it is believed that the relationship between sensors 160 and 164 is closer.

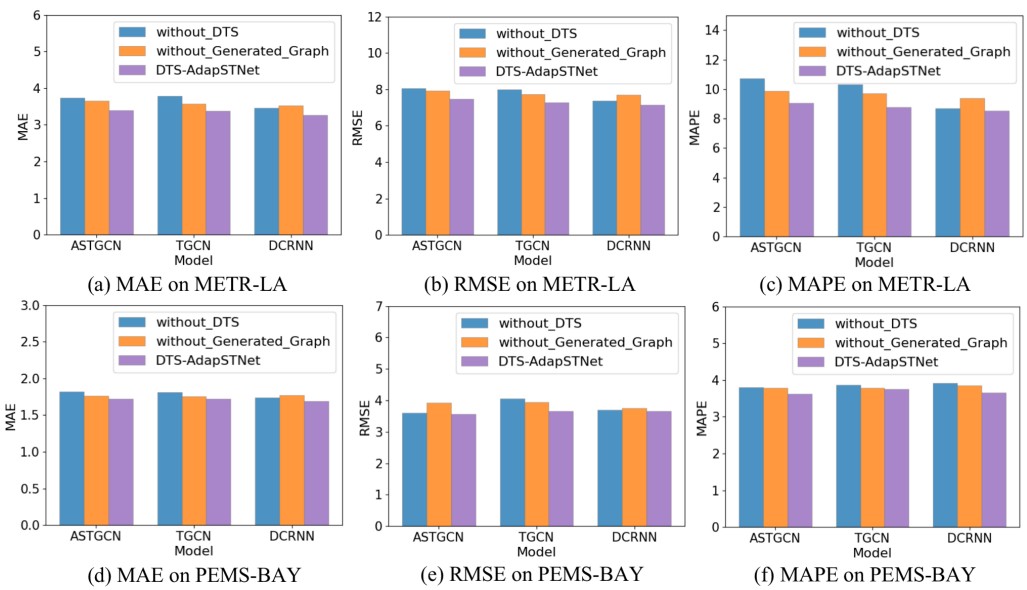

**Figure 9  Performance comparison of the tested models under different training construction.** (A, B, C) Performance comparison on METR-LA for traffic speed prediction with different models. (D, E, F) Performance comparison on PEMS-BAY for traffic speed prediction with different models.

As depicted in Fig. 10E, several observations can be made. Firstly, sensor 164 is situated downstream of sensor 160, indicating a strong correlation between the two sensors after model training. Secondly, there is a junction in the vicinity of sensor 164 where traffic can merge, and another junction near sensor 161 where traffic can flow out. Therefore, sensors 164 and 161 cannot be classified simply as part of the same segment. After model training, the color of square 2 becomes lighter, indicating a weakening of the spatial relationship between sensors 164 and 161. Finally, although all three sensors are located on the same road and are relatively close in distance, effective capturing of potential spatial relationships among them after training can enhance beneficial dependence relationships while weakening unfavorable ones. Consequently, new spatial relationships conducive to road speed prediction are obtained. It is evident that the proposed model demonstrates effectiveness in learning the spatial relationships among sensors on the same road, and endeavors to represent real road conditions accurately.

Regarding the spatial relationships among different roads, the proposed model also describes them through the spatial relationships of sensors located on different roads. Firstly, the two sensors located on the road 62 and the road 85 show a weak correlation in both the distance relationship matrix and the initialization relationship matrix, which are shown by square 3 in Figs. 10A and 10B. Secondly, after learning by the proposed model, it can be seen in the generated graph that the spatial relationship between the two sensors enhancs, which is shown by square 3 in Fig. 10C. This indicates that the two roads where the two sensors are located have a high similarity in traffic changes. Finally, to verify this, the real speed variation of the two roads in a day is plotted. As shown in Fig. 10D, it can be seen that road 62 and road 85 have almost identical speed curves. This further

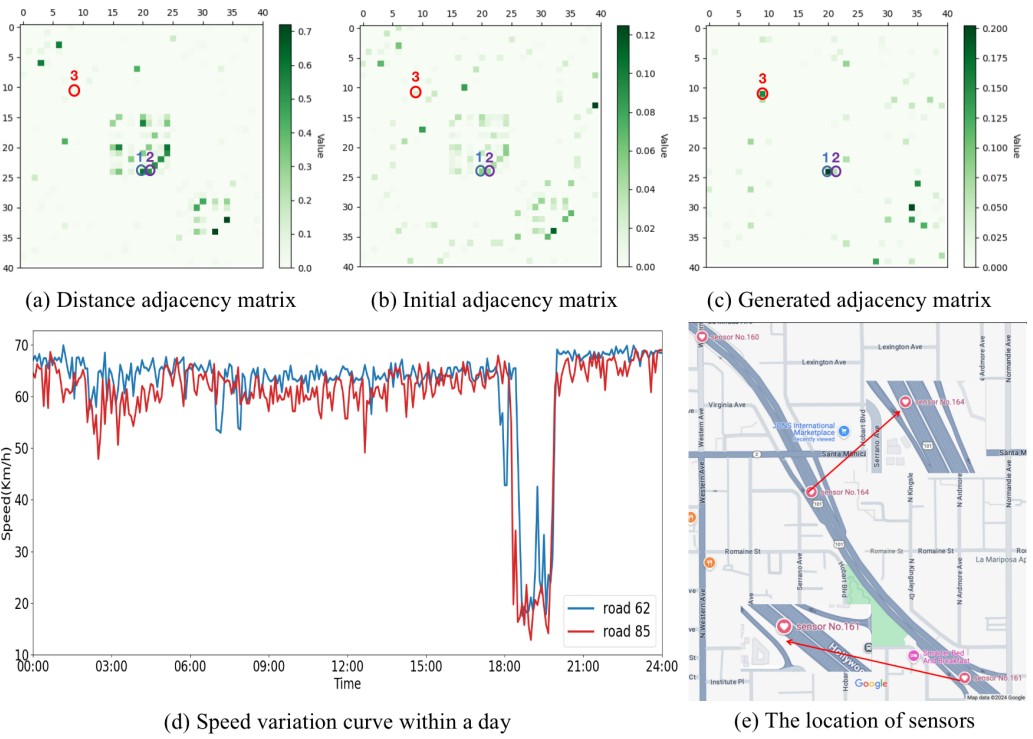

(a) Distance adjacency matrix    (b) Initial adjacency matrix    (c) Generated adjacency matrix

(d) Speed variation curve within a day    (e) The location of sensors

**Figure 10** Comparison of the matrices for analyzing the effect of adaptive graph generation module. Map data © 2024 Google.

demonstrates the importance of spatial relationships among the sensors represented by the generated graph in road network speed prediction.

In summary, the AMG-module is capable of effectively learning spatial relationships among sensors located on the same road as well as among sensors located on different roads. These spatial relationships contribute to making predictions more accurate.

## Improvement of the loss function

In order to investigate the contribution of the proposed loss function to the experimental outcomes, experiments are conducted on the METR-LA dataset using the DTS-AdapSTNet (ASTGCN) model. All other conditions remain unchanged, with only the loss function in the training process being modified. One version of the loss function is the general one, while the other is the optimized loss function proposed in this article. A comparison of the prediction results is performed, and the findings are presented in Table 4. It can be seen that after the proposed optimized loss function is used, the prediction performance of the model improves by 10.9%, 7.2% and 17.9% across all metrics, demonstrating the effectiveness of the proposed loss function.

Furthermore, in order to compare the impact of different loss functions on prediction results more intuitively, the alteration in speed for a specific road throughout a day in the test data is depicted in Fig. 11. Upon training with the proposed loss function, the resulting prediction curve aligns more closely with the actual data. Instead, when not

**Table 4** Performance comparison of DTS-AdapGCN(ASTGCN) model with different Loss function for speed prediction on METR-LA.

| Different loss function | MAE | RMSE | MAPE (%) |
|---|---|---|---|
| General loss function | 3.67 | 7.90 | 10.48 |
| Optimized loss functions | 3.27 | 7.33 | 8.6 |
| Improvement | 10.9% | 7.2% | 17.9% |

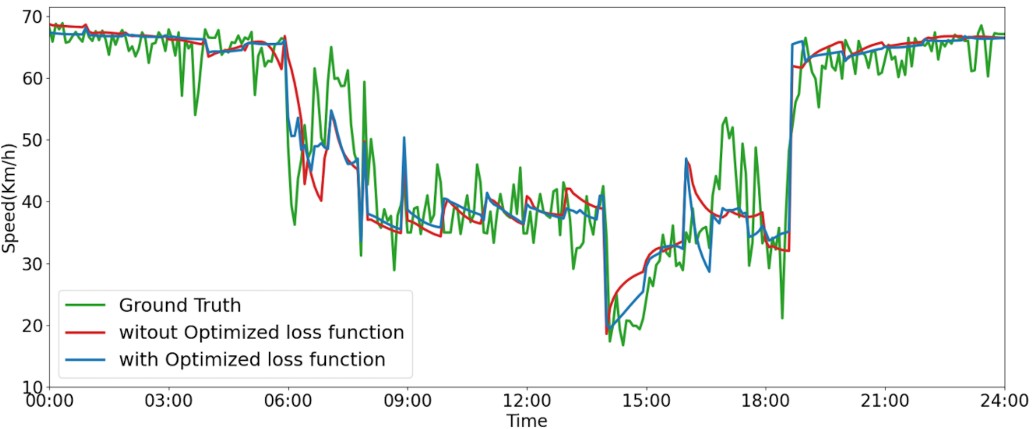

**Figure 11** Comparison of prediction curves on a road of the test data of METR-LA trained with different loss function.

utilizing the proposed loss function, the error is increased, illustrating the effectiveness and improvement introduced by the proposed loss function for road speed prediction.

## CONCLUSION

Capturing the potential spatial dependencies of roads in a traffic network to achieve accurate prediction poses a challenging problem. To address this challenge, an Adaptive Spatio-Temporal Graph Neural Network based on Multi-graph Fusion (DTS-AdapSTNet) is proposed in this article. In order to make more effective use of historical traffic data, firstly, DTS-AdapSTNet divides the roads in the road network and the sensors they contain carefully. The DTSRMG-module is used to capture the initial spatial dependencies among sensors, which are fused to generate an initial predefined matrix. Secondly, a novel AMG-module is proposed to learn the potential spatial dependencies adaptively. Specifically, the AMG-module and the prediction module are trained alternately in cycles, enabling the model to self-adjust. In addition, a loss function with good performance is designed in the process of model training. Furthermore, a fusion mechanism is used to fuse the learned matrices and produce the optimal adjacency matrix, thereby enhancing the accuracy of road traffic prediction. Finally, it is demonstrated through extensive experiments based on two real-world datasets that the proposed DTS-AdapSTNet outperforms other existing methods. Ablation experiments further confirm the effectiveness and contribution of each module in this model. Accurate prediction of roads in the traffic network is crucial for

urban travel, and the proposed method can assist intelligent transportation systems with route planning and management to make better decisions.

In the future, the proposed DTS-AdapSTNet can be improved in the following aspects. (1) External features that affect road traffic conditions (*e.g.*, weather, points of interest, emergencies, *etc.*) will be considered to enrich the DTSRMG-module proposed in this article. (2) The AMG-module will be modified so that it can train a dynamically changing adjacency matrix with real-time features. (3) The scalability of DTS-AdapSTNet in large-scale dynamic traffic environments requires further exploration. Although this article focuses on real-world datasets, the computational requirements for broader, city-wide implementations will need to be addressed in future work. Techniques such as model parallelization, distributed computing, and the use of more efficient graph learning algorithms will be explored to optimize the model's performance in large-scale systems. (4) The proposed module will predict the traffic conditions of urban roads more accurately using sensor data.

### Funding
This work was supported by the National Natural Science Foundation of China (grant number 61902069), the Natural Science Foundation of Fujian Province of China (grant number 2021J011068), the Research Initiation Fund Program of Fujian University of Technology (GY-S24002), and the Fujian Provincial Department of Science and Technology Industrial Guidance Project (grant number 2022H0025). The funders had no role in study design, data collection and analysis, decision to publish, or preparation of the manuscript.

### Grant Disclosures
The following grant information was disclosed by the authors:
The National Natural Science Foundation of China: 61902069.
Natural Science Foundation of Fujian Province of China: 2021J011068.
Research Initiation Fund Program of Fujian University of Technology: GY-S24002.
Fujian Provincial Department of Science and Technology Industrial Guidance Project: 2022H0025.

### Competing Interests
The authors declare there are no competing interests.

### Author Contributions
- Wenlong Shi conceived and designed the experiments, performed the experiments, performed the computation work, prepared figures and/or tables, authored or reviewed drafts of the article, and approved the final draft.
- Jing Zhang conceived and designed the experiments, performed the experiments, performed the computation work, authored or reviewed drafts of the article, and approved the final draft.

- Xiangxuan Zhong performed the experiments, performed the computation work, prepared figures and/or tables, and approved the final draft.
- Xiaoping Chen analyzed the data, prepared figures and/or tables, and approved the final draft.
- Xiucai Ye analyzed the data, prepared figures and/or tables, and approved the final draft.

## Data Availability

The two processed datasets, Metry_LA and PEMS_BAY, are available at Zenodo: Shi, W. (2024). Data for DTS-adapSTNet [Data set]. Zenodo. https://doi.org/10.5281/zenodo. 12520380. They both include training, validation, and testing sets.

The raw measurements are available in the Supplementary File.

## Supplemental Information

Supplemental information for this article can be found online at http://dx.doi.org/10.7717/ peerj-cs.2527#supplemental-information.

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
