# Peer review of "DTS-AdapSTNet: an adaptive spatiotemporal neural networks for traffic prediction with multi-graph fusion"

_PeerJ Computer Science, doi:10.7717/peerj-cs.2527_

## Round 0.1 · original submission · Minor Revisions

Dear authors,

Thank you for your paper. According to reviewers, your paper needs revision and we encourage you to address the minor concerns and criticisms of reviewers and resubmit your article once you have updated it accordingly.

Best wishes,

Reviewer 1 ·

Basic reporting

no comment

Experimental design

The experiments were conducted on two real public datasets, which provide a robust basis for evaluating the model's performance. The paper compares the proposed DTS-adapSTNet model with several existing models to demonstrate its effectiveness.

Validity of the findings

- The experimental results show that the proposed model outperforms the comparison models in terms of prediction accuracy. This indicates that the multi-graph fusion and two-stage alternating training structure effectively enhance the model's ability to capture spatiotemporal dependencies.
- The model's superior performance is consistent across different datasets, suggesting that the approach is robust and generalizable to various traffic prediction scenarios.
- The use of an improved loss function and a novel weighted fusion mechanism allows for better utilization of traffic data, leading to more accurate predictions.

Additional comments

- The experiments demonstrate that the model effectively addresses key challenges in traffic prediction, such as the reliance on single spatial relationships and limitations in end-to-end training methods.
- The improved accuracy and robustness of the model have practical implications for intelligent transportation systems, potentially leading to better traffic management and planning.

Reviewer 2 ·

Basic reporting

The paper presents DTS-adapSTNet, an innovative traffic prediction framework that leverages adaptive spatiotemporal graph neural networks combined with multi-graph fusion to enhance traffic forecasting accuracy. By addressing challenges such as the inadequate representation of spatial dependencies and limitations in existing end-to-end training methods, the model integrates multiple traffic data sources and employs a two-stage adaptive learning approach. Experimental results demonstrate that DTS-adapSTNet significantly outperforms baseline models in predicting traffic speed, achieving notable reductions in prediction errors across real-world datasets.

Experimental design

The use of two large real-world datasets is commendable, as it enhances the generalizability of the findings. However, details about the specific datasets (e.g., their size, characteristics, and the time period covered) should be included to better assess the robustness of the model.

Validity of the findings

The claim that DTS-adapSTNet outperforms baseline models in terms of MAE, RMSE, and MAPE is a strong indicator of its effectiveness.

The validity of the findings hinges on the quality of the datasets used for training and testing. Ensuring that the data is accurate, representative, and free from biases is crucial. Any issues related to data quality should be addressed to bolster the findings.

Additional comments

It is suggested to include the discussion of state-of-the-art spatio-temporal traffic prediction methods, particularly those utilizing self-supervised learning. The paper could benefit from a discussion on how self-supervised learning techniques, such as those used in "GPT-ST: generative pre-training of spatio-temporal graph neural networks," enhance model performance. Exploring how generative approaches can pre-train representations could provide insights into improving DTS-adapSTNet’s initial feature extraction and overall predictive capabilities.

It would be better to discuss the scalability of DTS-adapSTNet, including computational requirements and potential optimizations for large-scale implementations in dynamic traffic environments.

The architecture, while innovative, may introduce unnecessary complexity. Simplifying certain components or providing clearer justifications for each module's necessity could improve the model's interpretability.

Reviewer 3 ·

Basic reporting

- All acronyms and initialisms should be defined the first time they appear in the text. This should be in the abstract as well as the article.
- What is the novelty of the work compared to existing approaches? The author should justify the research question and the novelty of the work in the second section.
- The work requires extensive proofreading; many typos and grammatical errors exist.
- I cannot see that Figure 2 provides the motivations of the work. I recommend omitting Figure 2 since it adds nothing. The subsection of the motivation is enough.

Experimental design

- The work fits well with the journal, and it strongly matches its scope.
- What is the novelty of the work compared to existing approaches? The author should justify the research question and the novelty of the work in the second section.
- The methodology of the work is clear.

Validity of the findings

- The authors should define the mentioned performance metrics and justify their use.
- The experimental evaluation asses the proposed work, and the section is introduced well.

Additional comments

N/A

---

## Round 0.2 · accepted · Accept

Dear Authors,

Thank you for the revised paper. Your paper seems sufficiently improved and ready for publication.

Best wishes

Reviewer 2 ·

Basic reporting

The DTS-AdapSTNet framework introduces a new way to combine multiple traffic data sources, which helps improve the accuracy of traffic predictions. This approach addresses limitations found in existing models.

The proposed framework features an adaptive learning mechanism that allows it to adjust how it processes information based on changing traffic conditions.

Elaborate on how the distance, transfer, and same-road relationship matrices are computed and their significance in the overall model. Include the specific algorithms or formulas used for these calculations.

Provide more insight into how the adaptive learning mechanism operates. Explain the criteria for adjusting the learning direction and how this impacts model performance.

Experimental design

The inclusion of ablation studies allows for a detailed analysis of the individual components of the framework. This approach helps to identify which elements contribute most to the overall performance, strengthening the conclusions drawn from the experiments.

It would be helpful to discuss how specific features of the DTS-AdapSTNet framework contribute to its superior performance. For instance, explain the role of the multi-graph fusion approach and how it captures spatial-temporal dependencies more effectively than the baseline models.

Validity of the findings

The experimental findings demonstrate that the DTS-AdapSTNet framework significantly outperforms baseline models in traffic prediction accuracy. Using two large real-world datasets, the model achieved notable reductions in prediction errors, evidenced by improvements in key metrics such as Mean Absolute Error (MAE), Root Mean Square Error (RMSE), and Mean Absolute Percentage Error (MAPE).

Additional comments

N/A

Reviewer 3 ·

Basic reporting

The authors have addressed all comments; i have no more issues. I recommend publishing the work in its current state.

Experimental design

No more issues

Validity of the findings

No more issues

Additional comments

No more issues